# Beyond Marginals: Capturing Dependent Returns through Joint Moments in Distributional Reinforcement Learning

## Abstract

Distributional reinforcement learning (DRL) has emerged as a paradigm that aims to learn full distributions of returns under a policy, rather than only their expected values. The existing DRL algorithms learn the return distribution independently for each action at a state. However, we establish that in many environments, the returns for different actions at the same state are statistically dependent due to shared transition and reward structure, and that learning only per-action marginals discards information that is exploitable for secondary objectives. We formalize a joint Markov decision process (MDP) view that lifts an MDP into a partially-observable MDP whose hidden states encode coupled potential outcomes across actions, and we derive joint distributional Bellman equations together with a joint iterative policy evaluation (JIPE) scheme with convergence guarantees. We introduce a deep learning method that represents joint returns with Gaussian mixture models with optimality and convergence guarantees. Empirically, we first validate the JIPE scheme on MDPs with known correlation structure. Then, we illustrate the learned joint structure in control and Arcade Learning Environment tasks using neural networks. Together, these results demonstrate that modeling return dependencies yields accurate joint moments and joint distributions that can help interpretability and be used in deriving safe and cost-efficient policies.

## 1 Introduction

Reinforcement learning (RL) has long been utilized as a powerful framework for sequential decision-making problems where the interaction of the agent and the environment follows a Markov decision process (MDP). An MDP $\mathcal{M} = (\mathcal{S}, \varsigma_0, \mathcal{A}, R, P, \gamma)$ is a quintuple where $\mathcal{S}$ designates the state space, $\varsigma_0 \in \Delta(\mathcal{S})$ is the initial distribution of states, $\mathcal{A}$ is the set of actions that the agent may take, $R : \mathcal{S} \times \mathcal{A} \to \Delta(\mathbb{R})$ is a stochastic, real-valued reward function, $P : \mathcal{S} \times \mathcal{A} \to \Delta(\mathcal{S})$ is the transition kernel, and $0 < \gamma < 1$ is the discount factor (Puterman, 1994). In conventional RL, the learning objective is to find a policy with maximal expected return, which captures the agent's cumulative reward throughout its interaction. A policy $\pi$ for MDP $\mathcal{M}$ may be thought of as a decision rule $\pi : \mathcal{S} \to \mathcal{A}$.[1] To evaluate the merit of a given policy $\pi$, the expected return starting from a state-action pair $(s, a)$ $\mathbb{E}[Z^\pi(s, a)] := \mathbb{E}[\sum_{t=0}^{\infty} \gamma^t R(s_t, a_t) \mid s_0 = s, a_0 = a]$, also known as the Q-function or the state-action value function, may be considered. We remark that $Z^\pi(s, a)$ is the random variable (RV) which we will refer to as the *state-action return*. The objective in RL, then, is to find an *optimal policy* $\pi^*$ with an optimal state-action value function, i.e., to find $\pi^* \in \arg\max_{\pi \in \Pi} Q^\pi(s, a)$, for all $(s, a) \in \mathcal{S} \times \mathcal{A}$, where $\Pi$ denotes the set of all possible policies for $\mathcal{M}$. Famously, an optimal state-action value function $Q^*$ satisfies the *Bellman optimality equation* (Bellman, 1957) $Q^*(s, a) = \mathbb{E}[R(s, a)] + \gamma \mathbb{E}[\max_{a' \in \mathcal{A}} Q^*(s', a')]$, a premise which many RL algorithms have been built upon (Mnih et al., 2015; Sutton, 1988; Watkins & Dayan, 1992; Sutton, 1991; Rummery & Niranjan, 1994; Bradtke & Barto, 1996; Lagoudakis & Parr, 2003; Konda & Tsitsiklis, 1999; Hasselt et al., 2016; Wang et al., 2016).

---

[1] A celebrated result states that in the discounted, infinite-horizon setting, a *stationary* and *deterministic* optimal policy $\pi^*$ exists (Agarwal et al., 2019). We direct our attention to this case.

Figure 1: Joint return distributions learned by our method in the CartPole environment. On the left: The pole is perfectly balanced, the returns of both actions are perfectly correlated and the joint distribution is a ridge. On the right: The pole starts to lose balance to one side, the joint distribution becomes less degenerate. The curves on the edge of the plot show the two marginal distributions, which would have been learned by a conventional DRL method.

**Distributional RL.** More recently, a new paradigm called distributional RL (DRL) (Bellemare et al., 2017a) has emerged, based on the argument that reducing the return to its average value can obscure important aspects of uncertainty, variability, and risk, often leading to suboptimal exploration or brittle policies in stochastic settings. DRL augments the RL paradigm by modeling the full probability distribution over returns, capturing higher-order moments and tail behavior. This richer characterization aims to enable agents to both improve in performance, as well as in secondary objectives such as sample efficiency and policy robustness. All DRL methods up to now have been built around the tenet of estimating marginal return distributions for each action independently. Although different methods estimate different characterizations for these marginal distributions, at the end of the day, the entity they propose to model and estimate is some characterization of the $\theta$-parameterized marginal state-action return $Z_\theta(s, a)$ for a given state $s$ and for each action $a \in \mathcal{A}$.

## 1.1 MOTIVATION

In this paper, we argue for the theoretical interest in pursuing an alternative approach: It is only natural, we think, to be curious about the *joint distribution of these state-action returns* (see Figure 1). We argue that there is a nontrivial set of MDPs where, given a state $s$, there is dependence to be discovered between the returns of different actions. We present the following two motivating examples, after which a formal explanation follows:

**Example 1.** *Consider an MDP with bounded $\mathcal{S} \subset \mathbb{R}^2$, $\mathcal{A} = \{1, 2\}$, and with reward $R(s, a) := x_a$ for $a \in \mathcal{A}$, where $x \sim \mathcal{N}(s, \Sigma)$, and $\Sigma$ is a non-diagonal positive definite matrix. At any state $s$, the rewards $R(s, 1)$ and $R(s, 2)$ will be dependent RVs with covariance $\Sigma_{1,2}$. Because the return $Z^\pi(s, a)$ of any policy $\pi$ for any state-action pair is a weighted sum of such dependent rewards, the returns will also have a nontrivial joint distribution.*

**Example 2.** *Consider an MDP with $\mathcal{S} = \mathbb{R}$ and $\mathcal{A} = \{-1, 1\}$. Let $X$ be a Bernoulli RV. Let the next state be determined in terms of a state-action-dependent measurable function of $X$ as $S' = f(s, a, X) = s + a - 1$ if $X = 1$ and $s + a$ otherwise, so that $P(\cdot \mid s, a) := \text{Law}(S')$. The stochasticity of the transition dynamics of the environment is dependent on the RV $X$. Clearly, then, the next state RVs $S'_1 = f(s, -1, X)$ and $S'_2 = f(s, 1, X)$ will be dependent.*

In the previous example, $X$ might be thought of as modeling an environmental factor such as wind, pushing the agent leftward. Examples of such factors may include market fluctuations, a system-wide latency spikes, factors which simultaneously affect the results of all possible actions an agent could take at that moment. The fact that these two examples are specifically constructed to have dependencies should not give the impression that such dependencies do not arise in regular RL problems. The dependence of action returns is highly intrinsic even in the presence of a deterministic reward function in applications of RL, especially those involving function approximation (cf. Section 4.)

## 1.2 CONTRIBUTION

Our main contributions are:

- We show that action returns in many MDPs are statistically dependent due to shared transition and reward dynamics, motivating the need to move beyond independent per-action return distributions.

- We formalize a joint MDP perspective that lifts an MDP into a partially-observable MDP (POMDP) with hidden states encoding coupled potential outcomes across actions.

- We derive joint distributional Bellman equations and propose the Joint Iterative Policy Evaluation (JIPE) scheme, establishing convergence to joint means and second moments.

- We develop a deep learning approach that fits $K$-component Gaussian mixture models to represent joint return distributions, with guarantees on optimality and convergence.

- We validate our approach in synthetic MDPs with known correlation structure, demonstrating that JIPE accurately recovers joint moments.

- We extend the method to control and ALE tasks, showing that learned joint distributions improve interpretability and allow for safer and cost-efficient reinforcement learning policies.

## 2  JOINT DRL: A PRINCIPLED FRAMEWORK

### 2.1  PRINCIPLED MODELING OF CORRELATIONS VIA JOINT DRL

Having established the existence of return dependence, we develop a framework to study this phenomenon. Two standard assumptions from the literature follow (Bellemare et al., 2023; Sutton & Barto, 2018).

**Assumption 1.** *$\mathcal{A}$ is finite and $|\mathcal{A}| = N$. In the rest of the work, we will directly take $\mathcal{A} = [N]$ for ease of notation, so each action will be referred to by an integer $1 \leq n \leq N$.*

**Assumption 2.** *For all $(s, a)$, $r_{\min} \leq R(s, a) \leq r_{\max}$ almost surely for some $r_{\min}, r_{\max} \in \mathbb{R}$.*

In light of the examples of the previous section, we formalize our analysis with the following.

**Definition 1** (Joint MDP). *Let $\mathcal{M}$ be an MDP $(\mathcal{S}, \varsigma_0, \mathcal{A}, R, P, \gamma)$. For any $s \in \mathcal{S}$, let $C_P(s)$ be some coupling on $\mathcal{S}^N$ with marginals $\{P(\cdot \mid s, i)\}_{i=1}^N$ and $C_R(s)$ some coupling on $\mathbb{R}^N$ with marginals $\{R(s, i)\}_{i=1}^N$. Consider the POMDP $\mathcal{J} = (\mathcal{X}, \varsigma_0', \mathcal{A}, P', R', \Omega, O, \gamma)$. Here, $\mathcal{X} := \mathcal{S}^N \times \mathbb{R}^N$. We write a typical element $x \in \mathcal{X}$ as $x = (\mathbf{s}, \mathbf{r})$, where $\mathbf{s} = (s_1, \ldots, s_N)$ and $\mathbf{r} = (r_1, \ldots, r_N)$. $\varsigma_0'(x) := \varsigma_0(s) \times \delta(\mathbf{0})$ if $s_i = s$ for all $i \in [N]$ and $0$ otherwise, where $\mathbf{0}$ indicates a vector of $N$ zeros. In other words, the initial distribution over $\mathcal{X}$ only assigns nonzero probability to configurations which initialize all $N$ states in $\mathbf{s}$ at the same state, with all initial rewards being $0$. $P'(\cdot \mid x, a) := C_P(s_a) \times C_R(s_a)$, the product measure of the transition and reward couplings. $R'(x, a) := r_a$, deterministic. The observation space is $\Omega := \mathcal{S} \times \mathbb{R}$ and the observation kernel is $O(o \mid x, a) := \delta_{(s_a, r_a)}(o)$.*

At each decision time $t$ within the POMDP $\mathcal{J}$, the hidden state is a pair of vectors $x_t = (\mathbf{s}_t, \mathbf{r}_t)$, with $\mathbf{s}_t = (s_{t,1}, \ldots, s_{t,N}) \in \mathcal{S}^N$ and $\mathbf{r}_t = (r_{t,1}, \ldots, s_{r,N}) \in \mathbb{R}^N$. The $i^{\text{th}}$ entries $s_{t,i}$ and $r_{t,i}$ denote the next state and reward that would be obtained if action $i$ were to be taken from the current base state. After the agent selects $a_t$, the environment reveals $(s_{t,a_t}, r_{t,a_t})$ and the base state updates to $s_{t+1} = s_{t,a_t}$, and a fresh pair $(\mathbf{s}_{t+1}, \mathbf{r}_{t+1})$ is drawn at $s_{t+1}$ according to the specified couplings. For initialization, a state $s \sim \varsigma_0$ is sampled and $\mathbf{s}_0 = (s, \ldots, s)$ is set. $\mathbf{r}_0 = \mathbf{0}$ is set as a placeholder, since the reward at the initial state is a reward obtained before any actions have been played, and hence has no meaning and is unused. This representation is observationally equivalent to the original MDP $\mathcal{M}$: For any $(s, a)$, the revealed pair $(s_a, r_a)$ has exactly the same distribution as $(S', R)$ under the kernels $P(\cdot \mid s, a)$ and $R(s, a)$ of $\mathcal{M}$, but in addition, the POMDP's hidden state preserves the joint *counterfactual* outcomes across actions at each step. This makes it meaningful to learn joint statistics and to write joint Bellman relations, without altering the agent's observed interaction process.

**In practice, how do we model this joint MDP?** The following definitions formalize the vector-valued RV of joint returns whose distribution we aim to estimate.

**Definition 2.** *Let $Z^\pi(s, a)$ denote the state-action return of policy $\pi$ at $(s, a)$. Then, the $N$-variate joint return of policy $\pi$ at $s$ is defined as $Z^\pi(s) = [Z^\pi(s, 1), \ldots, Z^\pi(s, N)]^T$.*

**Definition 3.** *Let $\eta^\pi(s, a) = \text{Law}(Z^\pi(s, a))$. Then, the joint return distribution of policy $\pi$ at $s \in \mathcal{S}$ is a coupling of $\{\eta^\pi(s, i)\}_{i=1}^N$. Additionally, to denote the bivariate marginal distribution of $\eta^\pi(s)$ over the $i^{\text{th}}$ and $j^{\text{th}}$ dimensions, we use*

$$\eta^\pi(s; i, j) = \int_{z_{\bar{\mathbf{a}}}} \eta^\pi(s) \mathrm{d}z_{\bar{\mathbf{a}}}. \tag{1}$$

*The notation $\mathrm{d}z_{\bar{\mathbf{a}}}$ denotes that the integral is over dimensions $\mathcal{A}\backslash\{i,j\}$.*

We now provide an example for intuition on a problem we face in estimating joint distributions.

**Example 3.** *Suppose a simple scenario where we want to estimate a bivariate Gaussian distribution but only observe marginal samples. We observe $x_1, \ldots, x_N$ from the first and $y_1, \ldots, y_N$ from the second marginals, each in $\mathbb{R}$. We use the sample mean and variance to estimate the true parameters by $\hat{\mu}_1, \hat{\mu}_2, \hat{\sigma}_1^2$ and $\hat{\sigma}_2^2$. We can then estimate the joint distribution as $\mathcal{N}(\hat{\boldsymbol{\mu}}, \hat{\Sigma})$, where $\hat{\boldsymbol{\mu}} = [\hat{\mu}_1 \quad \hat{\mu}_2]^T$, and the marginal covariances are $\hat{\sigma}_1^2$ and $\hat{\sigma}_2^2$. The cross-covariance must then be $\rho\hat{\sigma}_1\hat{\sigma}_2$. However, we realize that although we can make educated estimates for the marginal statistics, it is impossible to do so for $\rho$ without observing joint samples from the bivariate distribution.*

In this work, we will be interested in learning the joint distribution $\eta^\pi(s)$ associated with a reference policy $\pi$, or certain statistical functionals that could aid us in inferring it. Classic RL is concerned with learning the mean $\mu^\pi(s) \in \mathbb{R}^{\mathcal{A}}$, i.e., the state-action value function $Q$. As we are interested in inferring the correlations, a natural functional to consider in addition to $\mu^\pi(s)$ is the $N \times N$ covariance matrix derived from $\eta^\pi(s)$ which we denote by $\Sigma^\pi(s)$.

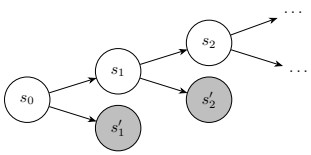

Figure 2: The state transitions shown as a tree. Starting at $s_0$, we store two possible next states $s_1$ and $s'_1$ reached by taking two actions. The next two states to be stored $s_2$ and $s'_2$ are only possible next states reachable from $s_1$. Next states reachable from $s'_1$ are not considered. This prevents the number of stored states from increasing exponentially.

However, as illustrated by Example 3, since any off-diagonal element $\Sigma^\pi(s)_{a,a'}$ of $\Sigma^\pi(s)$ relates knowledge about the joint returns of two actions at state $s$, the customary transition structure of $\tau := (s, a, r, s', a')$ will no longer suffice to estimate these elements. If we hope to learn a meaningful joint distribution of the returns of multiple actions at a state, we must change the structure of our saved and sampled experience replays to be $\tau^2 := (s, a_1, a_2, r_1, r_2, s'_1, s'_2, a'_1, a'_2)$, where $a_1$ and $a_2$ are two distinct actions that can potentially be played at state $s$, $r_1$ and $r_2$ the ensuing rewards, $s'_1$ and $s'_2$ the respective next states, and $a'_1$ and $a'_2$ the actions chosen by $\pi$ in the next states. Much like how, in the conventional DRL setting, we would expect the observation of the transition $\tau$ to lend us guidance in updating our estimate of $\eta^\pi(s, a)$, we would now expect to leverage the observation of $\tau^2$ to update our estimates of $\mu^\pi(s)_{a_1}, \mu^\pi(s)_{a_2}$, the diagonal covariance elements $\Sigma^\pi(s)_{a_1,a_1}, \Sigma^\pi(s)_{a_2,a_2}$ and finally the off-diagonal covariance elements $\Sigma^\pi(s)_{a_1,a_2}$ and $\Sigma^\pi(s)_{a_2,a_1}$. Obviously, this would result in updating our estimate of the bivariate marginal distribution $\eta^\pi(s; a_1, a_2)$. From now on, we will refer to such transition samples as $\tau^2$ as *joint transitions*.

In the formalism of Definition 1, to get access to joint transitions, we must change our observation space and kernel to allow us a peek into the joint structure. Letting $\Omega := \mathcal{S}^2 \times \mathbb{R}^2$ and $O(o \mid x, a_1) := \delta_{((s_{a_1}, s_{a_2}),(r_{a_1}, r_{a_2}))}(o)$ suffices. At any step of the joint MDP, the observation model lets us peek into the next state and reward of $a_1$ (dictated by a policy $\pi$), and those of one additional, counterfactual action $a_2$ that could have been played instead. This may very well be thought of theoretically as having access to an oracle which may be queried to obtain samples of joint transitions, partially uncovering the joint structure of the POMDP. This presumption can be thought of in the same light as oracle access to sample transitions already present in standard RL literature (Agarwal et al., 2019; Bellemare et al., 2023; Kearns & Singh, 1998), or access to counterfactual trajectories in explainable RL literature (Amitai et al., 2024).

**Remark 1.** *In the context of this work, access to joint transitions of the form $\tau^2$ is sufficient, since second moments relate information about pairs of actions. If one wants to model using distributions that require higher-order moments, it is not hard to generalize the results of the paper to this setting. For instance, for third moments, we would indeed require samples that give us information about triples of actions. We would then have to assume oracle access from the POMDP that gives us this type of information, i.e., we would need $\Omega := \mathcal{S}^3 \times \mathbb{R}^3$ and $O(o \mid x, a_1) := \delta_{(s_{a_1}, s_{a_2}, s_{a_3}),(r_{a_1}, r_{a_2}, r_{a_3})}(o)$. Access to such a model would give us access to joint transitions of form $\tau^3 := (s, a_1, a_2, a_3, r_1, r_2, r_3, s'_1, s'_2, s'_3, a'_1, a'_2, a'_3)$, relating the consequences of two additional, counterfactual actions. It is straightforward from this example to see how this would generalize to yet higher-order moments.*

**Can we obtain joint transitions from an MDP in practice?** Many applications of RL are increasingly relying on digital twin technologies, enabling a near-perfect simulation of reality. In many RL tasks, it is not implausible to assume having access to a perfect simulation of the system which allows taking an action, observing its consequences, rewinding the simulation to the previous state and then taking another action to observe its consequences. In common benchmark simulation environments that many RL works use, such as the ALE Atari environments or Gymnasium control environments, it is remarkably simple to achieve this effect by modifying only a handful lines of code. It is as simple as saving a temporary backup of the state of the random number generator(s) and the environment before taking an action, recording the next state and reward that follow, restoring the state of the random number generator(s) and the environment to the backed-up states, taking a different action and recording its next state, and rewards, and so on, for the required number of joint actions. Because all such rewards/next states are obtained under the same, common source of stochasticity, we may then view these as joint samples from the joint POMDP.

We do acknowledge that there will naturally be scenarios and applications where we will not be able to achieve this, or have a simulator at all. Even in these scenarios, we posit that there are possible workarounds. We refer the reader to Appendix I for a discussion.

With the discussion on joint transitions settled, we now introduce the joint Bellman equations.

**Definition 4.** *Let $Z^\pi(s)$ be the $N$-variate joint return under $\pi$ and $(S = s, A_1 = a_1, R_1, S'_1, A'_1, A_2 = a_2, R_2, S'_2, A'_2)$ a sample transition. The $2^{nd}$-order joint Bellman distributional equations are*

$$Z^\pi(s, a_1) \overset{D}{=} R_1 + \gamma Z^\pi(S'_1, A'_1),$$

$$(Z^\pi(s, a_1))^2 \overset{D}{=} (R_1 + \gamma Z^\pi(S'_1, A'_1))^2,$$

$$Z^\pi(s, a_1) \cdot Z^\pi(s, a_2) \overset{D}{=} (R_1 + \gamma Z^\pi(S'_1, A'_1)) \cdot (R_2 + \gamma Z^\pi(S'_2, A'_2)).$$

**Proposition 1.** *Let $(S = s, A_1 = a_1, R_1, S'_1, A'_1, A_2 = a_2, R_2, S'_2, A'_2)$ be a sample transition. The $2^{nd}$-order joint Bellman equations are*

$$\mathbb{E}[Z^\pi(s, a_1)] = \mathbb{E}[R_1 + \gamma Z^\pi(S'_1, A'_1) \mid S = s, A_1 = a_1],$$

$$\mathbb{E}[(Z^\pi(s, a_1))^2] = \mathbb{E}[(R_1 + \gamma Z^\pi(S'_1, A'_1))^2 \mid S = s, A_1 = a_1],$$

$$\mathbb{E}[Z^\pi(s, a_1) \cdot Z^\pi(s, a_2)] =$$
$$\mathbb{E}[(R_1 + \gamma\ Z^\pi(S'_1, A'_1)) \cdot (R_2 + \gamma Z^\pi(S'_2, A'_2)) \mid S = s, A_1 = a_1, A_2 = a_2],$$

*where $\mathbb{E}[\cdot]$ denotes the expectation with respect to the joint distribution over all RVs involved.*

Evidently, these equations provide us with consistency conditions that the first and second moments of the return $Z^\pi(s, a)$ must satisfy, in distribution and in expectation.

## 2.2 JOINT ITERATIVE POLICY EVALUATION (JIPE)

We can compactly represent the $2^{nd}$-order joint Bellman equations by defining a suitable operator. For each $(s, a)$, let $M_{(s,a)} \in \mathbb{R}^{N+1}$ be a vector that concatenates $\mathbb{E}[Z^\pi(s, a)]$ and $\mathbb{E}[(Z^\pi(s, a))^2]$ as its first and second coordinates and $\mathbb{E}[Z^\pi(s, a) \cdot Z^\pi(s, \tilde{a})]$, where $\tilde{a} \in \mathcal{A}, \tilde{a} \neq a$, as its last $N - 1$ coordinates. With this notation, let us define, for all $(s, a)$,

$$M_\mu(s, a) := M_{(s,a),1}, \quad M_\mu \in \mathbb{R}^{\mathcal{S} \times \mathcal{A}}$$

$$M_\sigma(s, a) := M_{(s,a),2}, \quad M_\sigma \in \mathbb{R}^{\mathcal{S} \times \mathcal{A}}$$

$$M_c(s, a) := \begin{bmatrix} M_{(s,a),3} & \cdots & M_{(s,a),N+1} \end{bmatrix}^T, \quad M_c \in \mathbb{R}^{(N-1) \times \mathcal{S} \times \mathcal{A}}$$

$$M := \begin{bmatrix} M_\mu^T & M_\sigma^T & M_c^T \end{bmatrix}^T, \quad M \in \mathbb{R}^{(N+1) \times \mathcal{S} \times \mathcal{A}}.$$

$M$ describes the collection of the means and the second moments of the $N$-variate joint return, collected by $M_\mu$ and $M_\sigma, M_c$, respectively. We can further represent the $2^{nd}$-order joint Bellman equations by the following 2nd-order $N$-variate joint Bellman operator

$$\mathcal{T}_{2,N}^\pi : \mathbb{R}^{(N+1) \times \mathcal{S} \times \mathcal{A}} \to \mathbb{R}^{(N+1) \times \mathcal{S} \times \mathcal{A}}, \qquad M = \mathcal{T}_{2,N}^\pi M.$$

We propose the following dynamic programming approach, the *joint iterative policy evaluation (JIPE)* scheme, which repeatedly applies the 2$^{\text{nd}}$-order joint Bellman operator $\mathcal{T}_{2,N}^{\pi}$

$$M^{k+1} = \mathcal{T}_{2,N}^{\pi} M^k, \quad M^0 \in \mathbb{R}^{(N+1) \times \mathcal{S} \times \mathcal{A}}. \tag{2}$$

Theorem 1 (proved in Appendix A) states the convergence of the scheme in (2). For simplicity of notation, the theorem is stated in terms of the uncentered second moment matrix, $\bar{\Sigma}^{\pi}(s)$, from which the covariance can be derived as $\Sigma^{\pi}(s) = \bar{\Sigma}^{\pi}(s) - \mu^{\pi}(s)\mu^{\pi}(s)^T$.

**Theorem 1** (Convergence of JIPE). *Suppose Assumptions 1 and 2 hold. Consider the JIPE scheme in (2). For any $s \in \mathcal{S}$, let $\mu^k(s)$ and $\bar{\Sigma}^k(s)$ denote the mean and the second moment matrix recovered from $M^k$. Then,*

$$\|\mu^k(s) - \mu^{\pi}(s)\|_{\infty} = \mathcal{O}\left(\gamma^k\right), \quad \|\bar{\Sigma}^k(s) - \bar{\Sigma}^{\pi}(s)\|_{\infty} = \mathcal{O}\left(\frac{\gamma^k}{1-\gamma}\right).$$

## 3 LEARNING OPTIMAL JOINT DISTRIBUTIONS VIA NEURAL NETWORKS

We now present an algorithmic approach to learning the joint distribution, leveraging the deep learning (DL) paradigm. We propose to model the state-action return as a Gaussian mixture model with $K$ components ($K$-GMM), whose parameters are estimated by a neural network with weights $\theta$. We note that Choi et al. (2019) and Zhang (2023) have previously suggested this approach, however, as usual, these works only consider the estimation of the marginal and not of the joint returns. For reasons such as reduced computational complexity and feasibility, we choose to only deal with homoscedastic $K$-GMMs, i.e., for a given $s \in \mathcal{S}$, $\Sigma_{\theta,i}(s) = \Sigma_{\theta}(s)$ for all $i \in [K]$.

We remind the reader of the discussion in Section 2 of we must rely on joint transitions to be able to learn the correlation structure of the joint returns. (See Figure 2 for a way to gather joint transitions without the number of states exponentially increasing.) We now propose the following distributional variant of the standard Q-learning algorithm, utilizing joint transitions: At each update step, for a sampled experience replay transition $\tau^2 := (s, a_1, a_2, r_1, r_2, s_1', s_2', a_1', a_2')$, we calculate the distributional temporal difference error between the current state's bivariate marginal return distribution $\eta_{\theta}(s; a_1, a_2)$, and a TD target distribution $\eta_{\omega}^*(s_1', s_2')$, which will be the distribution of an RV we denote by $\mathbf{r} + \gamma Z_{\omega}^*(s_1', s_2')$. In other words, we take our temporal difference error to be $\mathcal{L}\left(\eta_{\theta}(s; a_1, a_2), \eta_{\omega}^*(s_1', s_2')\right)$, which then gets used to update the neural network weights $\theta$ through backpropagation and stochastic gradient descent methods.[2] In theory, any statistical distance may be used for $\mathcal{L}$. To justify this choice, we state two theorems regarding the representation error and distributional convergence of GMMs in the Cramér distance $d_C$. We refer the reader to Appendix C for the details of the statements and the proofs, and to Appendix B for equivalent results in the Wasserstein distance.

**Theorem 2** (Representation error of $d_C$-optimal $K$-GMM). *Let $\eta^{\pi}(s)$ and $\hat{\eta}^{\pi}(s)$ denote the $N$-variate joint return distribution of policy $\pi$ and the distribution of its $d_C$-optimal $K$-GMM approximation, respectively. Then, under Assumption 2, it holds that for any $s \in \mathcal{S}$,*

$$d_C(\eta^{\pi}(s), \hat{\eta}^{\pi}(s)) \leq \frac{\sqrt{N}(r_{\max} - r_{\min})}{(1-\gamma)K^{1/N}}.$$

**Theorem 3** (Distributional convergence of 1-GMMs in $d_C$). *Instate the notation and hypotheses of Theorem 1. Let $\eta^k(s) = \mathcal{N}(\boldsymbol{\mu}^k(s), \Sigma^k(s))$, where $\Sigma^k(s)$ is the covariance derived from the uncentered matrix of second moments $\bar{\Sigma}^k(s)$. Let $\hat{\eta}^{\pi}(s) = \mathcal{N}(\boldsymbol{\mu}^{\pi}(s), \Sigma^{\pi}(s))$ be the 1-GMM approximation of the true return distribution $\eta^{\pi}(s)$. Assume that $\Sigma^{\pi}(s)$ is positive definite for all $s \in \mathcal{S}$. Then, for any $s \in \mathcal{S}$,*

$$d_C(\hat{\eta}^{\pi}(s), \eta^k(s)) = \mathcal{O}\left(\sqrt{N}\gamma^k + \frac{N\gamma^k}{(1-\gamma)\sqrt{\lambda_{\min}(\Sigma^{\pi}(s))}}\right).$$

The nature of $\eta_{\omega}^*(s_1', s_2') = \text{Law}(\mathbf{r} + \gamma Z_{\omega}^*(s_1', s_2'))$ must now be specified. This distribution resembles the familiar TD target of both classic RL and conventional DRL settings, but due to its

---

[2]We remind that the target RV is calculated with a separate set of parameters $\omega$ (as opposed to $\theta$), the parameters of the so-called *target network* (Mnih et al., 2015).

Table 1: SRB joint moments, calculated analytically and estimated by JIPE.

| | True | JIPE | $\|\Delta\|_\infty$ |
|---|---|---|---|
| $\boldsymbol{\mu}^T$ | $\begin{bmatrix} 1.8 & 2.0 \end{bmatrix}$ | $\begin{bmatrix} 1.8 & 2.0 \end{bmatrix}$ | $1.849 \times 10^{-12}$ |
| Corr | $\begin{bmatrix} 1.000 & 0.942 \\ 0.942 & 1.000 \end{bmatrix}$ | $\begin{bmatrix} 1.000 & 0.942 \\ 0.942 & 1.000 \end{bmatrix}$ | $4.441 \times 10^{-16}$ |

multivariate nature, some clarifications must be made. In truth, $\eta_\omega^*(s_1', s_2')$ is a coupling: It is a bivariate joint distribution whose univariate marginal distributions are the TD target distributions for $\{\eta_\theta(s, a_i)\}_{i=1}^2$, i.e., $\mathrm{Law}(r_i + \gamma Z_\omega(s_i', a_i^*))$, where $a_i^* \in \arg\max_{a' \in \mathcal{A}} \mathbb{E}[Z_\omega(s; a')]$. Going back to GMM terminology, the $i^{\text{th}}$ univariate marginal dimension of $\eta_\omega^*(s_1', s_2')$ is a $K$-GMM with mixing coefficients $\rho_\omega(s_i')$ and means $r_i + \gamma \mu_{\omega,k}(s_i', a_i^*)$.

We have specified the mixing coefficients and the means of the TD target $\eta_\omega^*(s_1', s_2')$, and only the covariance remains. Let us refer to the covariance matrix as $\Sigma_\omega(s_1', s_2')$. Given our previous logic for constructing the coupling target distribution, our covariance matrix must now satisfy $\Sigma_{\omega,i,j}(s_1', s_2') = \mathrm{cov}(r_i + \gamma Z_\omega(s_i', a_i^*), r_j + \gamma Z_\omega(s_j'; a_j^*))$ as a sample-based estimate of the true covariance $\mathrm{cov}(R(s, a_i) + \gamma Z(S_i', a_i^*), R(s, a_j) + \gamma Z(S_j', a_j^*))$.

We remark that with the provision that the TD target distribution $\eta_\omega^*(s_1', s_2')$ must be a coupling of the TD target distributions of the two univariate marginal distributions $\eta_\theta(s, a_1)$ and $\eta_\theta(s, a_2)$, it must, at its most general form, be the distribution of a $K^2$-GMM. Letting $(k_1, k_2) \in [K]^2$ index the $K^2$ components of the target mixture, and referring back to the homoscedasticity assumption mentioned in the beginning of the section, we finally have

$$\eta_\omega^*(s_1', s_2') = \sum_{k_1=1}^K \sum_{k_2=1}^K (\rho_{\omega,k_1}(s_1') \cdot \rho_{\omega,k_2}(s_2')) \, \mathcal{N}\left( \begin{bmatrix} r_1 + \gamma\mu_{\omega,k_1}(s_1', a_1^*) \\ r_2 + \gamma\mu_{\omega,k_2}(s_2', a_2^*) \end{bmatrix}, \Sigma_\omega(s_1', s_2') \right).$$

In practice, this implies that at each update step, we are fitting a $K$-GMM to a $K^2$-GMM. This might be envisioned as distilling the most prominent features of the $K^2$-GMM down to a $K$-GMM, keeping the model size reasonably bounded at all times. Notably, in the case of 1-GMMs, both $\eta_\theta(s; a_1, a_2)$ and $\eta_\omega^*(s_1', s_2')$ have the same number of components.

## 4 EXPERIMENTAL RESULTS

### 4.1 JOINT ITERATIVE POLICY EVALUATION (JIPE)

We report two minimal MDPs that manifest correlated returns.

**Shared-Randomness Bandit (SRB).** A one-state, two-action MDP with reward $R_t \sim \mathcal{N}(\mu_r, \Sigma_r)$, in the spirit of Example 1. The shared Gaussian draw induces dependence between the two actions' rewards. We set $\mu_r = \begin{bmatrix} 0.0 & 0.2 \end{bmatrix}^T$. The variance of the first action is 0.8, the second action is 1.0 and the covariance of the two actions is 0.6. The evaluated policy plays action 2 for all time steps.

**Windy Gridworld (WGW).** A 3x3 grid-world environment with a leftward gust of wind, present with probability $p = 0.35$, akin to Example 2. The wind perturbs the transition dynamics irrespective of the chosen action, pushing the agent one cell to the left in addition to the action chosen. The agent starts in the bottom-left cell. The goal cell is absorbing and only the actions that land on the goal cell produce reward 1, all other state-action pairs produce zero reward. The evaluated policy is presented in Figure 3. For each setup we evaluate a fixed policy using the JIPE scheme to compute the means, and the covariance matrix. We derive closed-form ground truth values for the moments and observe precise agreement in terms of maximum absolute distance

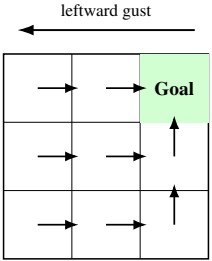

Figure 3: Deterministic policy evaluated in the WGW environment with leftward gust. The gust is shared each time step between all actions.

Table 2: WGW joint moments in the starting cell for actions `RIGHT`, `LEFT`, `UP`, `DOWN`.

| | True | | | | JIPE | | | | $\|\Delta\|_\infty$ |
|---|---|---|---|---|---|---|---|---|---|
| $\boldsymbol{\mu}^T$ | $\begin{bmatrix} 0.771 & 0.732 & 0.792 & 0.732 \end{bmatrix}$ | | | | $\begin{bmatrix} 0.771 & 0.732 & 0.792 & 0.732 \end{bmatrix}$ | | | | $2.004 \times 10^{-6}$ |
| Corr | $\begin{bmatrix} 1.000 & 0.833 & 0.866 & 0.833 \\ 0.833 & 1.000 & 0.866 & 1.000 \\ 0.866 & 0.866 & 1.000 & 0.866 \\ 0.833 & 1.000 & 0.866 & 1.000 \end{bmatrix}$ | | | | $\begin{bmatrix} 1.000 & 0.833 & 0.866 & 0.833 \\ 0.833 & 1.000 & 0.866 & 1.000 \\ 0.866 & 0.866 & 1.000 & 0.866 \\ 0.833 & 1.000 & 0.866 & 1.000 \end{bmatrix}$ | | | | $1.612 \times 10^{-4}$ |

Table 3: Scores achieved by unsafe policy $\pi_u$ and Markowitz policies $\pi_M$ with varying $\lambda$ values.

| Policy | Mean | Std. dev. | Min. | Max. | CVaR$_{0.1}$ | CVaR$_{0.05}$ |
|---|---|---|---|---|---|---|
| $\pi_u$ | $-213.4$ | $102.4$ | $-690.7$ | $\mathbf{-11.2}$ | $-415.95$ | $-455.64$ |
| $\pi_M$ with $\lambda = 0.01$ | $\mathbf{-118.7}$ | $101.1$ | $-724.0$ | $-15.0$ | $-337.64$ | $-388.74$ |
| $\pi_M$ with $\lambda = 10.0$ | $-127.1$ | $\mathbf{99.4}$ | $\mathbf{-568.3}$ | $-16.5$ | $\mathbf{-335.55}$ | $\mathbf{-383.34}$ |

within 20 iterations. The results are presented in Tables 1 and 2. These results
validate that the JIPE scheme recovers the moments implied by the coupled dynamics and rewards.

## 4.2 SAFETY THROUGH COVARIANCE ESTIMATION

To demonstrate how the estimated covariance matrices might be leveraged for safe RL, we use the
Cliff Walking environment (Sutton & Barto, 2018) (Figure 4) with a high slipping probability of $0.5$.
Consider the naive, performance-oriented but unsafe policy $\pi_u$ of walking straight along the edge of
the cliff to the goal state, shown in the figure in red. $\pi_u$ has the potential of reaching the goal state
within the least amount of steps. However, the high probability of slipping down into the cliff and
incurring a catastrophic negative consequences introduces a large amount of variance to its returns.

We first evaluate this policy through the JIPE scheme.
We then propose to use the mean-variance selection strat-
egy (Markowitz, 1952) in the context of portfolio op-
timization to derive a safer stochastic policy which in-
corporates information about the covariances. At state
$s$, we use a solution $\pi_M$ of the quadratic problem
$\max_{\pi \in \Delta_N} \pi^T \mu^{\pi_u}(s) - \lambda \pi^T \Sigma^{\pi_u}(s)\pi$ as a stochastic pol-
icy, where $\lambda$ strikes a balance between performance and
risk-aversion. This process can alternatively be viewed
as a post-hoc one-step policy improvement step, which,
given the first and second moments of a policy $\pi_u$, re-

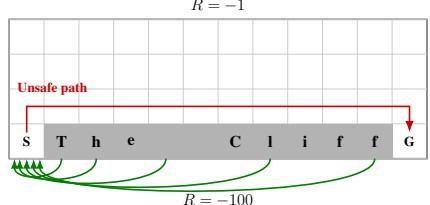

Figure 4: Cliff Walking environment
with evaluated unsafe policy outlined.

turns a policy $\pi_M$, improved in the sense of safety. We simulate the environment for 2500 test
episodes with the unsafe $\pi_u$ and with the Markowitz policy $\pi_M$ found with varying values of $\lambda$. The
results are in Table 3. The average score achieved by Markowitz policies are greatly improved over
$\pi_u$ for all values of $\lambda$.

## 4.3 INTERPRETABILITY THROUGH COVARIANCE ESTIMATION

We now showcase several DL applications where the learned covariances contribute to interpreting
the agent's policy and provide context about the environment state.

Firstly, we present the estimated joint distributions of near-optimal returns for two states from the
CartPole environment after 50 thousand frames using the DL framework of Section 3, with a 1-
GMM. Because this is an environment with $N = 2$, we are able to plot the joint distribution of
returns. Figure 1 shows these distributions for the given frames. The degenerate ridge structure
observed on the left, in the case of a bivariate Gaussian distribution, is observed when the correlation
coefficient satisfies $|\rho| \approx 1$. This indicates that the two are extremely (positively or negatively)

Table 4: Scores and costs of $\pi^*$ vs. $\pi_c$ across environments.

| Env. | Scores (mean $\pm$ sem) | | Costs (mean $\pm$ sem) | | $\pi_c$ vs. $\pi^*$ (%) | |
| | $\pi^*$ | $\pi_c$ | $\pi^*$ | $\pi_c$ | $\Delta$Score | $\Delta$Cost |
|---|---|---|---|---|---|---|
| Atlantis | 912204.0$\pm$10121.5 | **957336.0$\pm$12404.8** | 20483.2$\pm$158.3 | **16496.1$\pm$145.5** | **+4.9** | **-19.5** |
| Battle Zone | **29480.0 $\pm$ 1281.4** | 27440.0 $\pm$ 915.8 | 3243.1 $\pm$105.6 | **2946.7 $\pm$119.7** | -7.0 | **-9.2** |
| Boxing | 100.0 $\pm$ 0.0 | 100.0 $\pm$ 0.0 | 895.9 $\pm$ 1.7 | **799.5 $\pm$ 5.9** | +0.0 | **-10.8** |
| Pong | 21.0 $\pm$ 0.0 | 21.0 $\pm$ 0.0 | 2293.4 $\pm$ 13.8 | **722.3 $\pm$ 19.6** | +0.0 | **-68.5** |
| Video Pinball | **494829.5$\pm$11064.1** | 470557.1$\pm$22251.8 | 42018.4$\pm$118.3 | **37480.0$\pm$353.6** | -4.9 | **-11.2** |

correlated. We anticipate extreme correlation when the pole is perfectly balanced and stable, as the system is in near-complete symmetry and we also expect this symmetry in the covariance matrix, which is indeed the case here.

In the domain of the Arcade Learning Environment (ALE) (Bellemare et al., 2012), we present Figure 5, which shows three correlation matrices belonging to a near-optimal return distribution of Pong after 50 million training frames. The stark differences between the correlation matrices allow us to interpret the state of the environment and the decisions of the agent. On the left is a *noncritical state*. The game has just started, the ball is heading towards the opponent, there is no urgency to take any action as the agent has not observed how the ball will be heading towards them. The corresponding correlation matrix shows that the returns of actions are almost completely uncorrelated. In the middle are two *critical states*. The ball is heading toward the agent and the agent must now start taking the correct actions to return it. The matrix shows clear correlations and inverse correlations between the returns of actions, as taking incorrect actions at this point may lead to conceding. On the right is a *post-critical state*. By this point, the agent has taken the correct actions and has full belief that they have returned the ball with a perfect shot. Knowing that they have already scored, any actions taken while they wait have no effect on the outcome of the point. All actions after this point are perfectly correlated.

## 4.4 Cost-Efficiency through Covariance Estimation

With the intuition of the previous section, we propose a heuristic for the criticality of states. The observations and discussion on Figure 5 lead us to propose the (normalized) *effective rank* (Roy & Vetterli, 2007), $\mathrm{erank}(\Sigma) = \frac{\exp\left(-\sum_{i=1}^{N} p_i \log p_i\right)-1}{N-1}$, where $p_i = \frac{\lambda_i}{\sum_{j=1}^{N} \lambda_j}$ and $\lambda_i$ are the eigenvalues of matrix $\Sigma$. We observe from the figure that in noncritical states, there is small correlation between the returns of any two actions, and the correlation matrix is close to the identity. The correlation matrix is full rank, and $\mathrm{erank}(\Sigma) \approx 1$. Similarly, we observe that in post-critical states, there is very strong positive correlation between any two actions and the entries of the correlation matrix are all close to 1. The correlation matrix is a rank-one matrix, and $\mathrm{erank}(\Sigma) \approx 0$. These two observations lead us to consider the critical states as those which satisfy $\delta < \mathrm{erank}(\Sigma) < 1-\delta$ for some threshold parameter $\delta$. To test this hypothesis through the lens of cost-efficiency, we assign every ALE action an energy cost. The action NOOP, which corresponds to doing nothing, has zero cost. Simple actions such as UP, LEFT, FIRE cost 1 energy. Composites of two actions such as UPFIRE cost 2 energy. Composites of three actions such as UPLEFTFIRE cost 3 energy. We compare the policies $\pi^*$ and $\pi_c$, where $\pi^*$ is the near-optimal policy learned through the methodology of Section 3 after 50 million training frames, and $\pi_c$ is the cost-efficient policy which follows $\pi^*$ but ignores the dictated action at noncritical states and takes NOOP instead. We set the threshold parameter $\delta = 0.005$ for Atlantis and Battle Zone, 0.01 for Boxing, 0.015 for Pong and 0.0125 for Video Pinball. We present the average scores achieved and energy costs incurred by the two policies in these environments over 25 test episodes in Table 4. $\pi_c$ leads to significant reduction in energy cost with zero or small degradation in score, and even slight improvement in score in the case of Atlantis. It is perhaps also of note that the scores achieved in Atlantis beat those of C51 and Rainbow (Hessel et al., 2018), as reported in Figure 14 and Table 6 in the respective works.

**Remark 2.** *We would like to highlight that first moment indifference is not the same as outcome irrelevance. The value function can dangerously obscure the underlying risk structure as demonstrated by the following example: Consider a state $s$ where $\mu(s, a_1) = \mu(s, a_2) = 0.5$. An agent solely considering the first moment sees indifference and might simply play NOOP to save on cost. Let us con-*

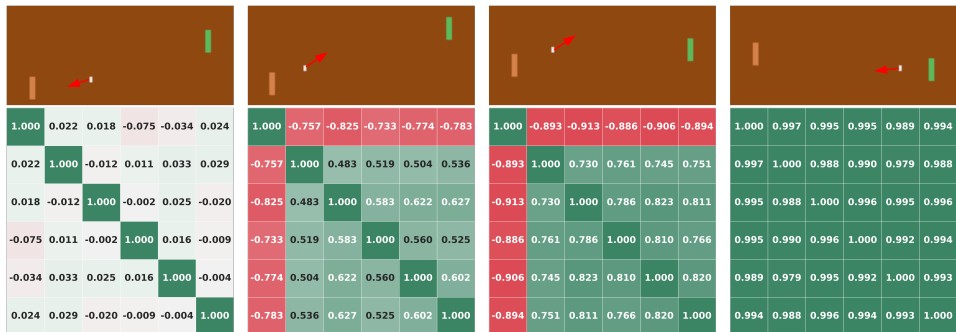

Figure 5: Four examples of covariance matrices of a near-optimal return at the shown states of Pong. The arrows are added by the authors to provide context as to where the ball is headed. The effective ranks are respectively 0.998, 0.397, 0.194, and 0.009.

*sider two cases: (i)* $Z(s, a_1) = Z(s, a_2) = 0.5$ *with probability 1. Here,* $\sigma_1^2 = \sigma_2^2 = 0$, $\rho_{12} = +1.0$ *and* $\mathrm{erank}(\Sigma) = 1$. *The actions are truly irrelevant and playing* NOOP *is a valid choice. (ii)* $Z(s, a_1)$ *is 0 or 1 with equal probability.* $Z(s, a_2) = Z(s, a_1)$ *with probability* $\epsilon$ *and* $Z(s, a_2) = 1 - Z(s, a_1)$ *with* $1 - \epsilon$. *Here,* $\sigma_1^2 = \sigma_2^2 = 1/4$, $\rho_{12} = \frac{1}{4}(1 - 2\epsilon)$. $\mathrm{erank}(\Sigma) = \exp(-(1 - \epsilon)\log(1 - \epsilon) - \epsilon \log \epsilon$ *is a continuous function of* $\epsilon$, *taking every value in* $[0, 1]$ *as* $\epsilon$ *sweeps from 0 to 1/2. Then, for any choice of* $\delta \in (0, 1)$, *an* $\epsilon$ *may be chosen such that* $\delta < \mathrm{erank}(\Sigma) < 1 - \delta$, *deeming the state critical. An agent only considering first moments cannot discriminate between these two cases, but our method using* $\mathrm{erank}(\Sigma)$ *can.*

## 5 CONCLUSION

We argued that dependencies between the returns of actions are intrinsic in many MDPs and developed a principled way to capture them by learning joint return distributions. We cast the problem as a POMDP whose hidden states store coupled potential outcomes across actions, derived joint Bellman equations and the JIPE scheme with convergence guarantees to the joint mean and second moments. We proposed a DL method that fits $K$-GMMs to estimate optimal joint return distributions. Empirical results on environments with known correlations and the proposed DL method on control and ALE tasks showed that the approach recovers accurate moments which may be used for safe, interpretable and cost-efficient RL. We envision that future research directions include estimating joint distributions as couplings of existing DRL methods and extensions to continuous action spaces.

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

## A  PROOF OF THEOREM 1

We first state a simple lemma which is used to derive the convergence results.

**Lemma 1.** *Consider two non-negative sequences $a_k$ and $b_k$. Assume $a_k \leq \gamma^k a_0$ and $b_{k+1} \leq a_0 B \gamma^k + \gamma^2 b_k$ for some $B > 0$ and $\gamma \in [0, 1)$. Then, $b_k \leq \gamma^{2k} b_0 + \frac{a_0 B \gamma^k}{1-\gamma}$.*

*Proof.* We proceed by unrolling the recurrence

$$
\begin{aligned}
b_{k+1} &\leq \gamma^2 b_k + a_0 B \gamma^k, \\
&\leq \gamma^2 (\gamma^2 b_{k-1} + a_0 B \gamma^k) + a_0 B \gamma^k \\
&= \gamma^4 b_{k-1} + a_0 B \gamma^{k+1} + a_0 B \gamma^k.
\end{aligned}
$$

Thus, by induction we have

$$
b_{k+1} \leq \gamma^{2(k+1)} b_0 + a_0 B \sum_{j=0}^{k} \gamma^{2(k-j)} \gamma^{j+1} = \gamma^{2(k+1)} b_0 + a_0 B \sum_{j=0}^{k} \gamma^{2(k-j)} \gamma^j.
$$

Using a change of variable $i = k - j$ we can calculate the second geometric sum:

$$
\sum_{j=0}^{k} \gamma^{2(k-j)} \gamma^j = \sum_{i=0}^{k} \gamma^{2i} \gamma^{k-i} = \gamma^k \sum_{i=0}^{k} \gamma^i = \gamma^k \cdot \frac{1 - \gamma^{k+1}}{1 - \gamma}.
$$

Using this in the previous equation furnishes the proof. $\qquad\square$

We now state the proof of the main result. We re-state the theorem for convenience.

**Theorem 4** (Convergence of $N$-variate joint iterative policy evaluation). *Suppose Assumptions 1 and 2 hold. Consider the $N$-variate joint iterative policy evaluation scheme in (2). For any $s \in \mathcal{S}$, let $\mu^k(s)$ and $\bar{\Sigma}^k(s)$ denote the mean and the uncentered matrix of second moments recovered from $M^k$. Then,*

$$
\|\mu^k(s) - \mu^\pi(s)\|_\infty = \mathcal{O}\left(\gamma^k\right), \quad \|\bar{\Sigma}^k(s) - \bar{\Sigma}^\pi(s)\|_\infty = \mathcal{O}\left(\frac{\gamma^k}{1-\gamma}\right).
$$

*Proof.* We adopt and strengthen an argument from Chapter 8 in Bellemare et al. (2023). We will first define the following semi-norms

$$
\begin{aligned}
\|M\|_{\infty,\mu} &= \sup_{(s,a)} |M_\mu(s,a)| \\
\|M\|_{\infty,\sigma} &= \sup_{(s,a)} |M_\sigma(s,a)| \\
\|M\|_{\infty,c} &= \sup_{(s,a,j)} |M_c(s,a)_j|
\end{aligned}
$$

Next, we demonstrate that the second-order $N$-variate joint Bellman operator $\mathcal{T}_{2,N}^\pi$ is a contraction with respect to $\|\cdot\|_{\infty,\mu}$ with constant $\gamma$. To see this, we remark that by the definition of $M_\mu$, $M$, and $\|\cdot\|_{\infty,\mu}$ we have that

$$
(\mathcal{T}_{2,N}^\pi M)_\mu = \mathcal{T}^\pi M_\mu,
$$

where $\mathcal{T}^\pi \mathbb{R}^{\mathcal{S} \times \mathcal{A}} \to \mathbb{R}^{\mathcal{S} \times \mathcal{A}}$ is the usual Bellman operator. Furthermore, note that $\|M\|_{\infty,\mu} = \|M_\mu\|_\infty$. Thus,

$$
\begin{aligned}
\|\mathcal{T}_{2,N}^\pi M - \mathcal{T}_{2,N}^\pi M'\|_{\infty,\mu} &= \|(\mathcal{T}_{2,N}^\pi M)_\mu - (\mathcal{T}_{2,N}^\pi M')_\mu\|_\infty \\
&= \|\mathcal{T}^\pi M_\mu - \mathcal{T}^\pi M'_\mu\|_\infty \\
&\leq \gamma \|M_\mu - M'_\mu\|_\infty \\
&= \gamma \|M - M'\|_{\infty,\mu}
\end{aligned}
$$

where we used the $\gamma$-contraction of $\mathcal{T}^\pi$ with respect to $\|\cdot\|_\infty$. Now recall, by linear convergence of the regular Bellman update $M_\mu^{k+1} = \mathcal{T}^\pi M_\mu^k$, we have

$$\|M^k - M^\pi\|_{\infty,\mu} = \|M_\mu^k - M_\mu^\pi\|_\infty \leq \gamma^k \|M_\mu^0 - M_\mu^\pi\|_\infty = \gamma^k \|M^0 - M^\pi\|_{\infty,\mu}.$$

This result establishes that the iterative policy evaluation scheme in (2) which repeatedly applies the second order $N$-variate joint Bellman operator $\mathcal{T}_{2,N}^\pi$ converges linearly to the mean of the $N$-variate joint return distribution $\eta^\pi(s)$.

To prove the rest of the statement, recall that for any $(s,a)$, by Assumption 2, $|\mathbb{E}[R(s,a)]| \leq \max\{|r_{\min}|, |r_{\max}|\} \leq B$ and $|\mathbb{E}[R(s,a)^2]| \leq \max\{|r_{\min}|^2, |r_{\max}|^2\} \leq B$ for some $B > 0$. Furthermore, by the definition of $M_\mu$, $M_\sigma$, $M$, $\|\cdot\|_{\infty,\mu}$, and $\|\cdot\|_{\infty,\sigma}$, for all $(s,a)$, we have

$$|(\mathcal{T}_{2,N}^\pi M)(s,a)_2 - (\mathcal{T}_{2,N}^\pi M')(s,a)_2| \leq 2B\gamma \left| \sum_{(s',a')\in\mathcal{S}\times\mathcal{A}} P(s' \mid s,a)\pi(a' \mid s')(M - M')(s',a')_1 \right|$$

$$+ \gamma^2 \left| \sum_{(s',a')\in\mathcal{S}\times\mathcal{A}} P(s' \mid s,a)\pi(a' \mid s')(M - M')(s',a')_2 \right|$$

$$\leq 2B\gamma\|M_\mu - M_\mu'\|_\infty + \gamma^2\|M_\sigma - M_\sigma'\|_\infty$$

$$= 2B\gamma\|M - M'\|_{\infty,\mu} + \gamma^2\|M - M'\|_{\infty,\sigma}$$

Hence,

$$\|\mathcal{T}_{2,N}^\pi M - \mathcal{T}_{2,N}^\pi M'\|_{\infty,\sigma} \leq 2B\gamma\|M - M'\|_{\infty,\mu} + \gamma^2\|M - M'\|_{\infty,\sigma}.$$

Similarly, we can establish a recursive inequality for the cross covariance $M_c$. In particular, for all $(s,a,j) \in \mathcal{S} \times \mathcal{A} \times \{3,\ldots,N+1\}$,

$$|(\mathcal{T}_{2,N}^\pi M)(s,a)_j - (\mathcal{T}_{2,N}^\pi M')(s,a)_j| \leq B\gamma \left| \sum_{(s_1',a_1')\in\mathcal{S}\times\mathcal{A}} P(s_1' \mid s,a)\pi(a_1' \mid s_1')(M - M')(s_1',a_1')_1 \right|$$

$$+ B\gamma \left| \sum_{(s_2',a_2')\in\mathcal{S}\times\mathcal{A}} P(s_2' \mid s,a_j)\pi(a_2' \mid s_2')(M - M')(s_2',a_2')_1 \right|$$

$$+ \gamma^2 \left| \sum_{(s',a')\in\mathcal{S}\times\mathcal{A}} P(s' \mid s,a)\pi(a' \mid s')(M - M')(s',a')_j \right|$$

$$\leq 2B\gamma\|M_\mu - M_\mu'\|_\infty + \gamma^2\|M_c - M_c'\|_\infty$$

$$= 2B\gamma\|M - M'\|_{\infty,\mu} + \gamma^2\|M - M'\|_{\infty,c}$$

where $a_j$ denotes the action used to calculate the cross covariance term for $(s,a)$ which is stored in $M_c(s,a)_j$, and we use the definition of the joint MDP, notably the fact that $P'(\cdot \mid x,a) := C_P(s_a) \times C_R(s_a)$, to bound the term in the bound (that is, the next state transition is dictated by $a$, not $a_j$). Hence,

$$\|\mathcal{T}_{2,N}^\pi M - \mathcal{T}_{2,N}^\pi M'\|_{\infty,c} \leq 2B\gamma\|M - M'\|_{\infty,\mu} + \gamma^2\|M - M'\|_{\infty,c}.$$

Thus, by invoking Lemma 1, one can readily establish

$$\|M^k - M^\pi\|_{\infty,\sigma} \leq \gamma^{2k}\|M^0 - M^\pi\|_{\infty,\sigma} + \frac{2\|M^0 - M^\pi\|_{\infty,\mu}B\gamma^k}{1-\gamma}$$

$$\|M^k - M^\pi\|_{\infty,c} \leq \gamma^{2k}\|M^0 - M^\pi\|_{\infty,c} + \frac{2\|M^0 - M^\pi\|_{\infty,\mu}B\gamma^k}{1-\gamma}$$

These results establish that the iterative policy evaluation scheme in (2) which repeatedly applies the 2nd order $N$-variate joint Bellman operator $\mathcal{T}_{2,N}^\pi$ converges linearly to the second moment (shifted covariance) of the $N$-variate joint return distribution $\eta^\pi(s)$. $\qquad\square$

# B  PROPERTIES OF $K$-GMMS UNDER THE WASSERSTEIN-2 DISTANCE

We first state a simple result, which is immediate from Assumption 2.

**Proposition 2** (Boundedness of return). *Under Assumption 2, $Z^\pi(s, a) \in \left[r_{\min}/(1-\gamma), r_{\max}/(1-\gamma)\right]$ almost surely. Furthermore, $Z^\pi(s) \in \left[r_{\min}/(1-\gamma), r_{\max}/(1-\gamma)\right]^N$ almost surely.*

Next, let us recall the definition of the $W_2$ distance and its important properties.

**Definition 5** (Wasserstein-2 distance). *Consider $\mathbb{R}^N$ with the Euclidean distance as the metric. Let $p$ and $q$ be two probability measures on $\mathbb{R}^N$ with bounded second moment, i.e.,*

$$\int_{\mathbb{R}^N} \|x\|^2 dp(x) < \infty, \qquad \int_{\mathbb{R}^N} \|x\|^2 dq(x) < \infty.$$

*Then, the Wasserstein-2 distance between $p$ and $q$ is defined as*

$$W_2(p, q) := \inf_{\alpha \in \Gamma(p,q)} \sqrt{\int_{(x,y)} \|x - y\|^2 d\alpha(x, y)},$$

*where $\Gamma(p, q)$ is the set of all couplings of $p$ and $q$.*

**Proposition 3** (Properties of $W_2$). *Consider two random variables $Z_p$ and $Z_q$ with distributions $p$ and $q$, respectively. With the notation $W_2(p, q) = W_2(Z_p, Z_q)$, the following holds*

- *1-homogeneity and regularity:*

$$W_2(X + \gamma Z_p, X + \gamma Z_q) \le \gamma W_2(Z_p, Z_q),$$

  *for all $\gamma \in [0, 1)$ and for any independent random variable $X$.*

- *2-convexity:*

$$W_2^2(\lambda p + (1 - \lambda)\hat{p}, \lambda q + (1 - \lambda)\hat{q}) \le \lambda W_2^2(p, q) + (1 - \lambda)W_2^2(\hat{p}, \hat{q}),$$

  *for all $\lambda \in [0, 1]$ and probability measures $\hat{p}$ and $\hat{q}$.*

We now state two main results, regarding the representation error of optimal $K$-GMMs and the distributional convergence of 1-GMMs under $W_2$.

## B.1  REPRESENTATION ERROR

We first start by an intuitive definition of our optimality criterion of $K$-GMMs estimating a return $Z^\pi(s)$, in terms of its $W_2$ distance.

**Definition 6** ($W_2$-optimal $K$-GMM approximation). *Let $Z^\pi(s)$ and $\eta^\pi(s)$ denote the $N$-variate joint return random variable and its distribution, respectively, following policy $\pi$. The $W_2$-optimal $K$-GMM is a multivariate random variable $\hat{Z}^\pi(s)$ with the distribution $\hat{\eta}^\pi(s) = \sum_{i=1}^{K} \hat{\rho}_i(s)\mathcal{N}(\hat{\mu}_i(s), \hat{\Sigma}_i(s))$, which satisfies $W_2(\hat{\eta}^\pi(s), \eta^\pi(s)) \le W_2(\tilde{\eta}^\pi(s), \eta^\pi(s))$, for all $K$-GMM distributions $\tilde{\eta}^\pi(s)$. Here, $\hat{\rho}_i(s) \in \Delta_K$ are the mixture coefficients, $\hat{\mu}_i(s) \in \mathbb{R}^N$ are the mixture means, and $\hat{\Sigma}_i(s) \in \mathbb{S}_+^N$ are the mixture covariance matrices, where $\mathbb{S}_+^N$ denotes the space of real-valued $N \times N$ positive definite matrices.*

As stated, this definition establishes the optimality criterion for any $K$-GMM. We argue, however, that when one restricts themselves to 1-GMMs, the optimal GMM is found to be parameterized by the true mean and covariance of $Z^\pi(s)$, i.e., $\mu^\pi(s)$ and $\Sigma^\pi(s)$.

**Proposition 4** ($W_2$-optimal 1-GMM approximation). *Let $\mu^\pi(s)$ and $\Sigma^\pi(s)$ denote the mean and covariance of the $N$-variate joint return $Z^\pi(s)$, respectively. Then, if $K = 1$, the $W_2$-optimal 1-GMM approximation of $Z^\pi(s)$ has distribution $\hat{\eta}^\pi(s) = \mathcal{N}(\mu^\pi(s), \Sigma^\pi(s))$.*

The previous definition and proposition characterize the optimal $K$-GMM and 1-GMM representations of $Z^\pi(s)$, respectively, but make no guarantees on the accuracy of these representations. We establish, in Theorem 5, a bound on the representation error incurred by optimal $K$-GMMs.

**Theorem 5** (Representation error of $W_2$-optimal $K$-GMM). *Let $\eta^\pi(s)$ and $\hat{\eta}^\pi(s)$ denote the $N$-variate joint return distribution of policy $\pi$ and the distribution of its $W_2$-optimal $K$-GMM approximation, respectively. Then, under Assumption 2, it holds that for any $s \in \mathcal{S}$,*

$$W_2(\eta^\pi(s), \hat{\eta}^\pi(s)) \leq \frac{\sqrt{N}(r_{\max} - r_{\min})}{(1 - \gamma)K^{1/N}}.$$

*Proof.* We adopt an argument from computational optimal transport (Peyré et al., 2019) and optimal quantization theory (Gruber, 2004).

Recall that under Assumption 2, $Z^\pi(s) \in [r_{\min}/(1-\gamma), r_{\max}/(1-\gamma)]^N$ almost surely by Proposition 2. Let us consider partitioning $[r_{\min}/(1-\gamma), r_{\max}/(1-\gamma)]^N$ into $K$ disjoint cubes $Q_i$, $i \in [K]$.

The side of each cube will be of length $\frac{r_{\max} - r_{\min}}{(1-\gamma)K^{1/N}}$. Furthermore, the volume of each cube will be $\left(\frac{r_{\max} - r_{\min}}{(1-\gamma)K^{1/N}}\right)^N$.

Let $\mathcal{C} = \{c_1, \ldots, c_K\}$ denote the center of these cubes. This helps us to define these cubes formally as

$$Q_i := \left\{ z \in \left[\frac{r_{\min}}{1 - \gamma}, \frac{r_{\max}}{1 - \gamma}\right]^N \mid \|z - c_i\| \leq \|z - c_j\|, \ \forall j \neq i, \ j \in [K] \right\}.$$

Furthermore, note that $\|z - c_i\| \leq \frac{\sqrt{N}(r_{\max} - r_{\min})}{2(1-\gamma)K^{1/N}}$ for all $z \in Q_i$.

Let $w_i = \eta^\pi(s)(Q_i)$ and note that $w_i \geq 0$ and $\sum_{i=1}^K w_i = 1$ as $\eta^\pi(s)$ is a valid probability measure on $\mathbb{R}^N$. Now, define the empirical measure

$$P_e(z) := \sum_{i=1}^K w_i \delta(z - c_i),$$

where $\delta(z)$ is the standard delta Dirac function. To upper bound $W_2(\eta^\pi(s), P_e)$, we define a non-optimal coupling $\alpha$ between $\eta^\pi(s)$ and $P_e$ as follows: For each $i \in [K]$, couple all the mass of $\eta^\pi(s)$ in cube $Q_i$ to the center point $c_i$. That is, let

$$\tilde{\alpha}(z, z') := \sum_{i=1}^K \mathbf{1}_{z \in Q_i} \cdot \delta(z' - c_i) \cdot \eta^\pi(s)(z).$$

Using this coupling, we have

$$\begin{aligned}
W_2^2(\eta^\pi(s), P_e) &= \left( \inf_{\alpha \in \Gamma(\eta^\pi(s), P_e)} \sqrt{\int_{z,z'} \|z - z'\|^2 d\alpha(z, z')} \right)^2 \\
&\leq \int_{z,z'} \|z - z'\|^2 d\tilde{\alpha}(z, z') \\
&= \sum_{i=1}^K \int_{Q_i} \|z - c_i\|^2 d\eta^\pi(s)(z) \\
&\leq \left( \frac{\sqrt{N}(r_{\max} - r_{\min})}{2(1 - \gamma)K^{1/N}} \right)^2 \sum_{i=1}^K \int_{Q_i} d\eta^\pi(s)(z) \\
&= \left( \frac{\sqrt{N}(r_{\max} - r_{\min})}{2(1 - \gamma)K^{1/N}} \right)^2 \sum_{i=1}^K w_i \\
&= \left( \frac{\sqrt{N}(r_{\max} - r_{\min})}{2(1 - \gamma)K^{1/N}} \right)^2.
\end{aligned}$$

Next, let us consider the following GMM:

$$\tilde{\eta}^\pi(s) = \sum_{i=1}^K w_i \mathcal{N}(c_i, \Sigma).$$

Recall that, by definition, $W_2(\hat{\eta}^\pi(s), \eta^\pi(s)) \leq W_2(\tilde{\eta}^\pi(s), \eta^\pi(s))$. Furthermore, as $W_2$ is a metric, by the triangle inequality we obtain

$$W_2(\hat{\eta}^\pi(s), \eta^\pi(s)) \leq W_2(\tilde{\eta}^\pi(s), \eta^\pi(s))$$
$$\leq W_2(\tilde{\eta}^\pi(s), P_e) + W_2(P_e, \eta^\pi(s))$$
$$\leq W_2(\tilde{\eta}^\pi(s), P_e) + \frac{\sqrt{N}(r_{\max} - r_{\min})}{2(1-\gamma)K^{1/N}}.$$

In what follows, we will set $\Sigma$ such that $W_2(\tilde{\eta}^\pi(s), P_e) \leq \frac{\sqrt{N}(r_{\max}-r_{\min})}{2(1-\gamma)K^{1/N}}$, thereby proving the stated bound.

By 2-convexity and regularity of $W_2$,

$$W_2^2(\tilde{\eta}^\pi(s), P_e) \leq \sum_{i=1}^{K} w_i W_2^2(\mathcal{N}(c_i, \Sigma), \delta(z - c_i))$$
$$= \sum_{i=1}^{K} w_i W_2^2(\mathcal{N}(0, \Sigma), \delta(z))$$
$$= W_2^2(\mathcal{N}(0, \Sigma), \delta(z)),$$

using $\sum_{i=1}^{K} w_i = 1$. Next, we will upper bound $W_2^2(\mathcal{N}(0, \Sigma), \delta(z))$ using the independent coupling

$$W_2^2(\mathcal{N}(0, \Sigma), \delta(z)) \leq \int_z \int_{z'} \|z - z'\|^2 \delta(z') \, \mathcal{N}(0, \Sigma)(z) dz dz'$$
$$= \int_z \|z\|^2 \mathcal{N}(0, \Sigma)(z) dz = \mathbb{E}_{Z \sim \mathcal{N}(0, \Sigma)} \|Z\|^2 = \mathrm{Tr}(\Sigma),$$

where we used the properties for the trace of a matrix and the linearity of the trace operator. Setting $\Sigma$ such that $\mathrm{Tr}(\Sigma) = \frac{\sqrt{N}(r_{\max}-r_{\min})}{2(1-\gamma)K^{1/N}}$ finishes the proof. $\qquad\square$

## B.2 DISTRIBUTIONAL CONVERGENCE

We now state the following result, establishing the distributional convergence in $W_2$ distance of 1-GMMs to the $W_2$-optimal 1-GMM, under the iterative evaluation scheme introduced in Section 2.2.

**Theorem 6** (Distributional convergence of 1-GMMs in $W_2$ distance)**.** *Instate the notation and hypotheses of Theorem 1. Let $\eta^k(s) = \mathcal{N}(\mu^k(s), \Sigma^k(s))$, where $\Sigma^k(s)$ is the covariance derived from the uncentered matrix of second moments $\bar{\Sigma}^k(s)$ as $\Sigma^k(s) = \bar{\Sigma}^k(s) - \mu^k(s)\mu^k(s)^T$. Then, $\eta^k(s)$ linearly converges to $\hat{\eta}^\pi(s) = \mathcal{N}(\mu^\pi(s), \Sigma^\pi(s))$, i.e., to the $W_2$-optimal 1-GMM approximation of the $N$-variate joint return distribution $\eta^\pi(s)$. That is, for any $s \in \mathcal{S}$,*

$$W_2(\hat{\eta}^\pi(s), \eta^k(s)) = \mathcal{O}\left(\sqrt{N}\gamma^k + \sqrt{\frac{N\gamma^k}{1-\gamma} \cdot \left(1 + \frac{\lambda_{\max}(\Sigma^\pi(s))}{\lambda_{\min}(\Sigma^\pi(s))}\right)}\right).$$

*where $\lambda_{\max}(\cdot)$ and $\lambda_{\min}(\cdot)$ denote the maximum and minimum eigenvalues of their argument.*

*Proof.* Let $\Sigma^k(s) = \Sigma^\pi(s) + \Delta^k$. Note that, without loss of generality, we can assume $\Delta^k$ is positive definite, otherwise we set $\Sigma^k(s) = \Sigma^\pi(s) - \Delta^k$. Recall that the $W_2$ distance between two multivariate Gaussian distributions is given by

$$W_2^2(\hat{\eta}^\pi(s), \eta^k(s)) = \|\mu^k(s) - \mu^\pi(s)\|_2^2 + \mathrm{Tr}\left(\Sigma^\pi(s) + \Sigma^k(s) - 2\left(\Sigma^\pi(s)^{1/2}\Sigma^k(s)\Sigma^\pi(s)^{1/2}\right)^{1/2}\right).$$

Theorem 1 establishes the linear convergence of the mean. Thus $\|\mu^k(s) - \mu^\pi(s)\|_2^2 = \mathcal{O}(N\gamma^{2k})$ using norm properties. On the other hand,

$$\left(\Sigma^\pi(s)^{1/2}\Sigma^k(s)\Sigma^\pi(s)^{1/2}\right)^{1/2} = \left(\Sigma^\pi(s)^{1/2}(\Sigma^\pi(s) + \Delta^k)\Sigma^\pi(s)^{1/2}\right)^{1/2}$$
$$= \left(\Sigma^\pi(s)^2 + \Sigma^\pi(s)^{1/2}\Delta^k\Sigma^\pi(s)^{1/2}\right)^{1/2}.$$

Since the matrix square root operator is monotone and analytic on the positive definite cone, its Fréchet derivative exists (Higham, 2008). Thus, we can write the first-order (Fréchet) Taylor expansion of the matrix square root function around $\Sigma^\pi(s)^2$ as

$$\left(\Sigma^\pi(s)^2 + \Sigma^\pi(s)^{1/2}\Delta^k\Sigma^\pi(s)^{1/2}\right)^{1/2} = \Sigma^\pi(s) + X + o(\|\Sigma^\pi(s)^{1/2}\Delta^k\Sigma^\pi(s)^{1/2}\|_F),$$

where $X$ the is unique solution to the Sylvester equation:

$$\mathcal{T}(X) := \Sigma^\pi(s)X + X\Sigma^\pi(s) = \Sigma^\pi(s)^{1/2}\Delta^k\Sigma^\pi(s)^{1/2},$$

and by the linear convergence of the covariance established by Theorem 1, we have $\|\Sigma^\pi(s)^{1/2}\Delta^k\Sigma^\pi(s)^{1/2}\|_F = \mathcal{O}(\frac{N\gamma^k\lambda_{\max}(\Sigma^\pi(s))}{1-\gamma})$.

Note that for any unitary invariant matrix norm, notably the Frobenius norm, the Ando-Hemmen inequality establishes the Lipschitz continuity of the matrix square root (see, e.g., Equation (1) in Del Moral & Niclas (2018)). In our application, the Lipschitz constant is strictly smaller than $\frac{1}{\lambda_{\min}(\Sigma^\pi(s))}$. Consequently, the linear Sylvester operator $\mathcal{T}$ is positive definite and invertible. Hence,

$$\|X\|_F = \|\mathcal{T}^{-1}(\Sigma^\pi(s)^{1/2}\Delta^k\Sigma^\pi(s)^{1/2})\|_F$$
$$\leq \frac{\|\Sigma^\pi(s)^{1/2}\Delta^k\Sigma^\pi(s)^{1/2}\|_F}{\lambda_{\min}(\Sigma^\pi(s))}$$
$$= \mathcal{O}\left(\frac{N\gamma^k\lambda_{\max}(\Sigma^\pi(s))}{(1-\gamma)\lambda_{\min}(\Sigma^\pi(s))}\right).$$

Consequently,

$$\left(\Sigma^\pi(s)^2 + \Sigma^\pi(s)^{1/2}\Delta^k\Sigma^\pi(s)^{1/2}\right)^{1/2} = \Sigma^\pi(s) + \mathcal{O}\left(\frac{N\gamma^k\lambda_{\max}(\Sigma^\pi(s))}{(1-\gamma)\lambda_{\min}(\Sigma^\pi(s))}\right),$$

and

$$\left|\text{Tr}\left(\Sigma^\pi(s) + \Sigma^k(s) - 2\left(\Sigma^\pi(s)^{1/2}\Sigma^k(s)\Sigma^\pi(s)^{1/2}\right)^{1/2}\right)\right| \leq \left|\text{Tr}\left(\Sigma^k(s) - \Sigma^\pi(s)\right)\right|$$
$$+ \mathcal{O}\left(\frac{N\gamma^k\lambda_{\max}(\Sigma^\pi(s))}{(1-\gamma)\lambda_{\min}(\Sigma^\pi(s))}\right).$$

Using the linear convergence of the covariance (and in particular, its diagonal) as established by Theorem 1, we have $\left|\text{Tr}\left(\Sigma^k(s) - \Sigma^\pi(s)\right)\right| = \mathcal{O}(\frac{N\gamma^k}{1-\gamma})$, using norm properties.

Leveraging all of our findings and using the inequality $\sqrt{a+b} \leq \sqrt{a} + \sqrt{b}$ for positive $a$ and $b$ yields

$$W_2(\hat{\eta}^\pi(s), \eta^k(s)) = \mathcal{O}\left(\sqrt{N}\gamma^k + \sqrt{\frac{N\gamma^k}{1-\gamma} \cdot \left(1 + \frac{\lambda_{\max}(\Sigma^\pi(s))}{\lambda_{\min}(\Sigma^\pi(s))}\right)}\right).$$

$\square$

## C  PROPERTIES OF K-GMMS UNDER THE CRAMÉR DISTANCE

We extend the analysis of the representation error of a GMM approximation to the Cramér distance, which is an alternative metric on the space of probability distributions.

Much like the results on the $W_2$ distance (Appendix B), the analysis is predicated on the foundational assumption regarding the bounded nature of the reward function, which in turn ensures that the return distribution has bounded support. (cf. Assumption 2 and 2.) Next, we provide the definition of the Cramér distance and list its essential properties that are instrumental to the proof.

**Definition 7** (Cramér distance). *Consider $\mathbb{R}^N$ with the Euclidean distance as the metric. Let $p$ and $q$ be two probability measures on $\mathbb{R}^N$. Let $X$ and $X'$ be independent random variables drawn from $p$, and $Y$ and $Y'$ be independent random variables drawn from $q$. The squared Cramér distance between $p$ and $q$ is defined as*

$$d_C^2(p,q) := 2\mathbb{E}\|X - Y\| - \mathbb{E}\|X - X'\| - \mathbb{E}\|Y - Y'\|.$$

**Proposition 5** (Properties of $d_C$). *Consider two random variables $Z_p$ and $Z_q$ with distributions $p$ and $q$, respectively. With the notation $d_C(p,q) = d_C(Z_p, Z_q)$, the following holds:*

- *Metric property: $d_C$ satisfies the properties of a metric, including the triangle inequality:*
  $$d_C(p,r) \leq d_C(p,q) + d_C(q,r).$$

- *Convexity: For probability measures $p, \hat{p}, q, \hat{q}$ and any $\lambda \in [0,1]$,*
  $$d_C^2(\lambda p + (1-\lambda)\hat{p}, \lambda q + (1-\lambda)\hat{q}) \leq \lambda d_C^2(p,q) + (1-\lambda)d_C^2(\hat{p}, \hat{q}).$$

- *Relation to expected norm: For a distribution $p$ and a Dirac measure $\delta_c$ at point $c$, $d_C$ is bounded by the expected Euclidean distance:*
  $$d_C(p, \delta_c) \leq \mathbb{E}_{X \sim p}\|X - c\|.$$

- *Invariance under translation: For any two $d$-dimensional random vectors $\mathbf{X}$ and $\mathbf{Y}$, and for any constant vector $\mathbf{c} \in \mathbb{R}^d$, the following equality holds:*
  $$d_C(\mathbf{X} + \mathbf{c}, \mathbf{Y} + \mathbf{c}) = d_C(\mathbf{X}, \mathbf{Y}).$$

## C.1 REPRESENTATION ERROR

We now present the main theorem concerning the representation error bound with respect to the Cramér distance.

**Theorem 7** (Representation error of $d_C$-optimal $K$-GMM). *Let $\eta^\pi(s)$ and $\hat{\eta}^\pi(s)$ denote the $N$-variate joint return distribution of policy $\pi$ and the distribution of its $d_C$-optimal $K$-GMM approximation, respectively. Then, under Assumption 2, it holds that for any $s \in \mathcal{S}$,*

$$d_C(\eta^\pi(s), \hat{\eta}^\pi(s)) \leq \frac{\sqrt{N}(r_{\max} - r_{\min})}{(1-\gamma)K^{1/N}}.$$

*Proof.* The structure of this proof is analogous to that of Theorem 5, adapting the arguments from the Wasserstein-2 distance to the Cramér distance.

From Proposition 2, we recall that the support of the return distribution $\eta^\pi(s)$ is the hypercube $\mathcal{H} = [\frac{r_{\min}}{1-\gamma}, \frac{r_{\max}}{1-\gamma}]^N$. We partition this hypercube into $K$ disjoint cubic cells $Q_i$ for $i \in [K]$, with centers denoted by $\mathcal{C} = \{c_1, \ldots, c_K\}$. The side length of each cube is $L = \frac{r_{\max} - r_{\min}}{(1-\gamma)K^{1/N}}$. For any point $z \in Q_i$, the Euclidean distance to its center $c_i$ is bounded by half the main diagonal of the cube:

$$\|z - c_i\| \leq \frac{\sqrt{N}L}{2} = \frac{\sqrt{N}(r_{\max} - r_{\min})}{2(1-\gamma)K^{1/N}}.$$

Let $w_i = \eta^\pi(s)(Q_i)$ be the probability mass of the true distribution within cell $Q_i$. We define an empirical measure $P_e$ composed of Dirac delta functions at the cell centers:

$$P_e(z) := \sum_{i=1}^{K} w_i \delta(z - c_i).$$

The Cramér distance is known to be upper-bounded by the Wasserstein-2 distance, i.e., $d_C(p,q) \leq W_2(p,q)$. We may therefore utilize the intermediate quantization error bound derived in the proof of Theorem 5. Specifically, it was established that

$$W_2(\eta^\pi(s), P_e) \leq \frac{\sqrt{N}(r_{\max} - r_{\min})}{2(1-\gamma)K^{1/N}}.$$

This directly implies a bound on the Cramér distance between the true distribution and the discrete approximation:

$$d_C(\eta^\pi(s), P_e) \leq W_2(\eta^\pi(s), P_e) \leq \frac{\sqrt{N}(r_{\max} - r_{\min})}{2(1-\gamma)K^{1/N}}. \tag{3}$$

Now, let $\hat{\eta}^\pi(s)$ be the $d_C$-optimal $K$-GMM approximation to $\eta^\pi(s)$, and consider an intermediate GMM, $\tilde{\eta}^\pi(s) = \sum_{i=1}^K w_i \mathcal{N}(c_i, \Sigma)$. By the optimality of $\hat{\eta}^\pi(s)$, we have $d_C(\hat{\eta}^\pi(s), \eta^\pi(s)) \leq d_C(\tilde{\eta}^\pi(s), \eta^\pi(s))$. The triangle inequality for $d_C$ yields:

$$
\begin{aligned}
d_C(\hat{\eta}^\pi(s), \eta^\pi(s)) &\leq d_C(\tilde{\eta}^\pi(s), \eta^\pi(s)) \\
&\leq d_C(\tilde{\eta}^\pi(s), P_e) + d_C(P_e, \eta^\pi(s)) \\
&\leq d_C(\tilde{\eta}^\pi(s), P_e) + \frac{\sqrt{N}(r_{\max} - r_{\min})}{2(1 - \gamma)K^{1/N}}.
\end{aligned}
$$

It remains to bound the term $d_C(\tilde{\eta}^\pi(s), P_e)$. Applying the convexity property of the squared Cramér distance from Proposition 5:

$$
\begin{aligned}
d_C^2(\tilde{\eta}^\pi(s), P_e) &= d_C^2\left( \sum_{i=1}^K w_i \mathcal{N}(c_i, \Sigma), \sum_{i=1}^K w_i \delta_{c_i} \right) \\
&\leq \sum_{i=1}^K w_i d_C^2(\mathcal{N}(c_i, \Sigma), \delta_{c_i}) \\
&= \sum_{i=1}^K w_i d_C^2(\mathcal{N}(0, \Sigma), \delta_0) = d_C^2(\mathcal{N}(0, \Sigma), \delta_0),
\end{aligned}
$$

where the final step uses the translation-invariance of the Cramér distance and the fact that $\sum w_i = 1$. Using a property from Proposition 5 and Jensen's inequality:

$$
d_C(\mathcal{N}(0, \Sigma), \delta_0) \leq \mathbb{E}_{Z \sim \mathcal{N}(0, \Sigma)} \|Z\| \leq \left( \mathbb{E} \|Z\|^2 \right)^{1/2}.
$$

For a random vector $Z \sim \mathcal{N}(0, \Sigma)$, $\mathbb{E}\|Z\|^2 = \mathrm{Tr}(\Sigma)$. Therefore, $d_C(\tilde{\eta}^\pi(s), P_e) \leq \sqrt{\mathrm{Tr}(\Sigma)}$.

We select the covariance matrix $\Sigma$ to match the bound from (3). Let $\Sigma = \sigma^2 I$, where $I$ is the identity matrix. We set $\sigma$ such that:

$$
\sqrt{\mathrm{Tr}(\sigma^2 I)} = \sqrt{N\sigma^2} = \sigma\sqrt{N} = \frac{\sqrt{N}(r_{\max} - r_{\min})}{2(1 - \gamma)K^{1/N}},
$$

which implies $\sigma = \frac{r_{\max} - r_{\min}}{2(1-\gamma)K^{1/N}}$. With this choice, $d_C(\tilde{\eta}^\pi(s), P_e) \leq \frac{\sqrt{N}(r_{\max}-r_{\min})}{2(1-\gamma)K^{1/N}}$.

Substituting this result into the main inequality completes the proof:

$$
\begin{aligned}
d_C(\hat{\eta}^\pi(s), \eta^\pi(s)) &\leq \frac{\sqrt{N}(r_{\max} - r_{\min})}{2(1 - \gamma)K^{1/N}} + \frac{\sqrt{N}(r_{\max} - r_{\min})}{2(1 - \gamma)K^{1/N}} \\
&= \frac{\sqrt{N}(r_{\max} - r_{\min})}{(1 - \gamma)K^{1/N}}.
\end{aligned}
$$

$\square$

## C.2 Distributional Convergence

We derive a convergence result for the iterative policy evaluation scheme when the target distribution is approximated by a single multivariate Gaussian distribution (a 1-GMM). The convergence is analyzed with respect to the Cramér distance, providing an analogue to the Wasserstein-2 distance result in Theorem 6.

We first establish two key properties of the Cramér distance between multivariate Gaussian distributions, which are instrumental for the main proof.

**Lemma 2.** *Let $\eta_1 = \mathcal{N}(\mu_1, \Sigma_1)$ and $\eta_2 = \mathcal{N}(\mu_2, \Sigma_2)$ be two non-degenerate multivariate Gaussian distributions on $\mathbb{R}^N$. The Cramér distance $d_C(\eta_1, \eta_2)$ satisfies the following inequalities:*

1. ***Shift property:*** *The $d_C$ distance between two Gaussian distributions with identical covariance matrices is bounded by the Euclidean distance between their means:*

$$
d_C(\mathcal{N}(\mu_1, \Sigma), \mathcal{N}(\mu_2, \Sigma)) \leq \|\mu_1 - \mu_2\|_2.
$$

2. **Covariance property:** *The squared $d_C$ distance between two zero-mean Gaussian distributions is bounded by the squared Frobenius norm of the difference of their matrix square roots:*

$$d_C^2(\mathcal{N}(0, \Sigma_1), \mathcal{N}(0, \Sigma_2)) \leq \|\Sigma_1^{1/2} - \Sigma_2^{1/2}\|_F^2.$$

*Proof.* The proof relies on the property that the Cramér distance is upper-bounded by the Wasserstein-2 distance, $d_C(p, q) \leq W_2(p, q)$. The $W_2$ distance for a specific coupling provides an upper bound on the true $W_2$ distance (which is the infimum over all couplings) and therefore also on the Cramér distance. We construct convenient couplings for both properties.

**Proof of the shift property.** Let $p = \mathcal{N}(\mu_1, \Sigma)$ and $q = \mathcal{N}(\mu_2, \Sigma)$. We construct a coupling of $(X, Y)$ by letting $Z \sim \mathcal{N}(0, I)$ and defining:

$$X = \mu_1 + \Sigma^{1/2}Z,$$
$$Y = \mu_2 + \Sigma^{1/2}Z.$$

By construction, $X \sim \mathcal{N}(\mu_1, \Sigma)$ and $Y \sim \mathcal{N}(\mu_2, \Sigma)$. The expected squared Euclidean distance is:

$$\mathbb{E}\|X - Y\|_2^2 = \mathbb{E}\|(\mu_1 + \Sigma^{1/2}Z) - (\mu_2 + \Sigma^{1/2}Z)\|_2^2$$
$$= \mathbb{E}\|\mu_1 - \mu_2\|_2^2 = \|\mu_1 - \mu_2\|_2^2.$$

Applying the upper bound $d_C(p, q)^2 \leq \mathbb{E}\|X - Y\|_2^2$ furnishes the proof of the first property.

**2. Proof of the covariance property.** Let $p = \mathcal{N}(0, \Sigma_1)$ and $q = \mathcal{N}(0, \Sigma_2)$. We construct a coupling of $(X, Y)$ by letting $Z \sim \mathcal{N}(0, I)$ and defining:

$$X = \Sigma_1^{1/2}Z,$$
$$Y = \Sigma_2^{1/2}Z.$$

This is a valid coupling where $X \sim \mathcal{N}(0, \Sigma_1)$ and $Y \sim \mathcal{N}(0, \Sigma_2)$. Let $\Delta = \Sigma_1^{1/2} - \Sigma_2^{1/2}$. The expected squared Euclidean distance is computed as follows:

$$\mathbb{E}\|X - Y\|_2^2 = \mathbb{E}\|(\Sigma_1^{1/2} - \Sigma_2^{1/2})Z\|_2^2 = \mathbb{E}\|\Delta Z\|_2^2$$
$$= \mathbb{E}[\text{Tr}((\Delta Z)^T(\Delta Z))] = \mathbb{E}[\text{Tr}(Z^T\Delta^T\Delta Z)]$$
$$= \mathbb{E}[\text{Tr}(\Delta^T\Delta ZZ^T)] = \text{Tr}(\Delta^T\Delta\mathbb{E}[ZZ^T]).$$

Since $Z \sim \mathcal{N}(0, I)$, its covariance matrix is the identity, $\mathbb{E}[ZZ^T] = I$. Thus,

$$\mathbb{E}\|X - Y\|_2^2 = \text{Tr}(\Delta^T\Delta) = \|\Delta\|_F^2 = \|\Sigma_1^{1/2} - \Sigma_2^{1/2}\|_F^2.$$

Applying the upper bound $d_C(p, q)^2 \leq \mathbb{E}\|X - Y\|_2^2$ completes the proof of the second property. $\square$

We now state the main theorem and its proof.

**Theorem 8** (Distributional convergence of 1-GMMs in $d_C$ distance). *Instate the notation and hypotheses of Theorem 1. Let $\eta^k(s) = \mathcal{N}(\mu^k(s), \Sigma^k(s))$, where $\Sigma^k(s)$ is the covariance derived from the uncentered matrix of second moments $\bar{\Sigma}^k(s)$. Let $\hat{\eta}^\pi(s) = \mathcal{N}(\mu^\pi(s), \Sigma^\pi(s))$ be the 1-GMM approximation of the true return distribution $\eta^\pi(s)$. Assume that $\Sigma^\pi(s)$ is positive definite for all $s \in \mathcal{S}$. Then, for any $s \in \mathcal{S}$,*

$$d_C(\hat{\eta}^\pi(s), \eta^k(s)) = \mathcal{O}\left(\sqrt{N}\gamma^k + \frac{N\gamma^k}{(1 - \gamma)\sqrt{\lambda_{\min}(\Sigma^\pi(s))}}\right).$$

*Proof.* Let $\eta^k(s) = \mathcal{N}(\mu^k, \Sigma^k)$ and $\hat{\eta}^\pi(s) = \mathcal{N}(\mu^\pi, \Sigma^\pi)$ for notational simplicity. We bound the Cramér distance by applying the triangle inequality with an intermediate distribution $\tilde{\eta}(s) = \mathcal{N}(\mu^\pi, \Sigma^k)$:

$$d_C(\hat{\eta}^\pi(s), \eta^k(s)) \leq d_C(\mathcal{N}(\mu^\pi, \Sigma^\pi), \mathcal{N}(\mu^\pi, \Sigma^k)) + d_C(\mathcal{N}(\mu^\pi, \Sigma^k), \mathcal{N}(\mu^k, \Sigma^k)). \quad (4)$$

We bound each of the two terms on the right-hand side separately.

For the second term in (4), which involves distributions with identical covariance, we use Lemma 2.1. This gives $d_C(\mathcal{N}(\mu^\pi, \Sigma^k), \mathcal{N}(\mu^k, \Sigma^k)) \leq \|\mu^\pi - \mu^k\|_2$. From Theorem 1, we have $\|\mu^k(s) - \mu^\pi(s)\|_\infty = \mathcal{O}(\gamma^k)$. Relating the infinity norm to the Euclidean norm yields:

$$\|\mu^k(s) - \mu^\pi(s)\|_2 \leq \sqrt{N}\|\mu^k(s) - \mu^\pi(s)\|_\infty = \mathcal{O}(\sqrt{N}\gamma^k).$$

For the first term in (4), which involves distributions with identical means, the translation-invariance of the Cramér distance and Lemma 2.2 imply

$$d_C^2(\mathcal{N}(\mu^\pi, \Sigma^\pi), \mathcal{N}(\mu^\pi, \Sigma^k)) = d_C^2(\mathcal{N}(0, \Sigma^\pi), \mathcal{N}(0, \Sigma^k)) \leq \|(\Sigma^\pi)^{1/2} - (\Sigma^k)^{1/2}\|_F^2.$$

The matrix square root function is Lipschitz continuous on the cone of positive definite matrices. As $\Sigma^k(s) \to \Sigma^\pi(s)$, we have the bound $\|(\Sigma^\pi)^{1/2} - (\Sigma^k)^{1/2}\|_F \leq \mathcal{O}\left(\frac{1}{\sqrt{\lambda_{\min}(\Sigma^\pi)}}\right)\|\Sigma^\pi - \Sigma^k\|_F$.

We bound the term $\|\Sigma^\pi(s) - \Sigma^k(s)\|_F$ by analyzing its components, $\Sigma(s) = \bar{\Sigma}(s) - \mu(s)\mu(s)^T$. The difference is $\Sigma^\pi - \Sigma^k = (\bar{\Sigma}^\pi - \bar{\Sigma}^k) - (\mu^\pi(\mu^\pi)^T - \mu^k(\mu^k)^T)$. By Theorem 1 and standard norm inequalities, the Frobenius norm of this difference is dominated by the convergence rate of the uncentered second moments, giving $\|\Sigma^\pi(s) - \Sigma^k(s)\|_F = \mathcal{O}\left(\frac{N\gamma^k}{1-\gamma}\right)$. Combining these results gives the bound for the covariance-related term:

$$d_C(\mathcal{N}(\mu^\pi, \Sigma^\pi), \mathcal{N}(\mu^\pi, \Sigma^k)) = \mathcal{O}\left(\frac{N\gamma^k}{(1-\gamma)\sqrt{\lambda_{\min}(\Sigma^\pi(s))}}\right).$$

Substituting the bounds for both terms back into the triangle inequality in (4) yields the final convergence rate:

$$d_C(\hat{\eta}^\pi(s), \eta^k(s)) = \mathcal{O}\left(\sqrt{N}\gamma^k\right) + \mathcal{O}\left(\frac{N\gamma^k}{(1-\gamma)\sqrt{\lambda_{\min}(\Sigma^\pi(s))}}\right).$$

This completes the proof. $\square$

# D  MARGINALIZATION IN GMMs

We take the time to discuss the relationship between *indexing by actions* and *marginalization* in $K$-GMMs, which is helpful to the exposition in Section 3. Note that $Z(s)$, unindexed by any action $a \in \mathcal{A}$, indicates the $N$-variate random variable of state-action returns for state $s$ and for all actions. We once again refer to the convention $\mathcal{A} = [N]$ established in Assumption 1, and indicate by $Z(s, i)$ the $i^{\text{th}}$ component of $Z(s)$:

$$Z(s) = [Z(s, 1) \quad Z(s, 2) \quad \ldots \quad Z(s, N)]^T.$$

Similarly, the marginal distribution of the $i^{\text{th}}$ component of $Z(s)$ may be expressed in terms of the joint distribution $\eta(s)$ as in (1), where we "marginalize out" every dimension except for the $i^{\text{th}}$ dimension through integration. Note that it is straightforward to extend this definition to indexing by multiple distinct actions, where, for instance, $Z(s; i, j)$ would indicate the bivariate joint random variable of state-action returns at state $s$ and for actions $i$ and $j \in [N]$, whose distribution $\eta(s; i, j)$ would be obtained by integrating over every dimension of $\eta(s)$ except the $i^{\text{th}}$ and $j^{\text{th}}$ dimensions.

Fortunately, with the choice of $N$-variate mixture of jointly-Gaussian random variables to model $Z(s)$, the integration in (1) becomes as simple as selecting relevant entries from the mean vectors $\mu_k(s)$ and from the covariance matrices $\Sigma_k(s)$ to parameterize yet another $K$-GMM distribution in $\mathbb{R}$. Indeed, if $\eta(s) = \sum_{k=1}^K \rho_k(s)\mathcal{N}(\mu_k(s), \Sigma_k(s))$, then,

$$\eta(s; j) := \sum_{k=1}^K \rho_k(s)\,\mathcal{N}\left((\mu_k(s))_j, \Sigma_k(s)_{j,j}\right).$$

Similarly, it is simple enough to extend this to multivariate marginal distributions of $Z(s)$. One only has to extract the multiple relevant entries from the mean vectors, and select the relevant sub-block

matrix out of the covariance matrices of the mixture. For instance, a marginalization of the $1^{\text{st}}$ and the $3^{\text{rd}}$ dimensions would simply be obtained by

$$
\Sigma_k(s) = \begin{bmatrix} \Sigma_k(s)_{1,1} & \Sigma_k(s)_{1,2} & \Sigma_k(s)_{1,3} & \cdots & \Sigma_k(s)_{1,N} \\ \Sigma_k(s)_{2,1} & \Sigma_k(s)_{2,2} & \Sigma_k(s)_{2,3} & \cdots & \Sigma_k(s)_{2,N} \\ \Sigma_k(s)_{3,1} & \Sigma_k(s)_{3,2} & \Sigma_k(s)_{3,3} & \cdots & \Sigma_k(s)_{3,N} \\ \vdots & \vdots & \vdots & \ddots & \vdots \\ \Sigma_k(s)_{N,1} & \Sigma_k(s)_{N,2} & \Sigma_k(s)_{N,3} & \cdots & \Sigma_k(s)_{N,N} \end{bmatrix}
$$

$$
\implies \Sigma_k(s;1,3) = \begin{bmatrix} \Sigma_k(s)_{1,1} & \Sigma_k(s)_{1,3} \\ \Sigma_k(s)_{3,1} & \Sigma_k(s)_{3,3} \end{bmatrix},
$$

and

$$
\mu_k(s) = \begin{bmatrix} (\mu_k(s))_1 & (\mu_k(s))_2 & (\mu_k(s))_3 & \cdots & (\mu_k(s))_N \end{bmatrix}^T
$$

$$
\implies \mu_k(s;1,3) = \begin{bmatrix} (\mu_k(s))_1 & (\mu_k(s))_3 \end{bmatrix}^T,
$$

for each $k \in [K]$, and hence,

$$
\eta(s;1,3) = \sum_{k=1}^{K} \rho_k(s) \, \mathcal{N}\left(\mu_k(s;1,3), \Sigma_k(s;1,3)\right),
$$

once again, the distribution of a $K$-GMM, this time in $\mathbb{R}^2$.

# E    RELATED WORK

A very early view of the DRL paradigm was first introduced by Morimura et al. (2010a;b), where the concept of the distributional Bellman equations were first laid out. These works, prior to the advent of deep learning methods, propose (non)parametric estimators for the modeling of the distribution of returns.

After the proliferation of deep learning methods in the context of reinforcement learning, and the success of DQN (Mnih et al., 2015), a sequence of DRL methods within this paradigm were proposed. Dabney et al. (2018a) propose a taxonomy of such methods, based on their two characteristics: How they parameterize the return distribution, and the distance metric they choose to optimize. We will adhere to this taxonomy in the following exposition. C51 (Bellemare et al., 2017a), which reinvigorated the field of DRL, and its extension Rainbow (Hessel et al., 2018) propose to model the return distribution of each state-action pair as a categorical distribution. In their case, a neural network produces a single categorical marginal distribution of $51$ parameters for each one of the $|\mathcal{A}|$ actions. They propose to use the Kullback-Leibler (KL) divergence as loss function. QR-DQN (Dabney et al., 2018b), IQN (Dabney et al., 2018a) and FQF (Yang et al., 2019) take a somewhat orthogonal approach and propose to model the inverse CDF, also known as the quantile function, with increasing levels of degrees of freedom, increasing the expressivity of the methods. They optimize the Huber quantile regression loss.

The most similar DRL method to this work is MoG-DQN, proposed by Choi et al. (2019). They propose to model the marginal return distributions using Gaussian mixture models (GMMs) and use the Jensen-Tsallis distance, i.e., the $\ell^2$ distance between the probability distributions, as loss function. In a similar vein, Zhang (2023) proposes the use of GMMs in RL, but proposes the optimization of the Cramér-2 distance instead, which we adopt in the experimental results of this work.

DRL methods which consider multivariate reward functions bear resemblance to our work. To name a few, the Bellman GAN model (Freirich et al., 2019) is proposed as a GAN-based approach to learn a deep generative model of the return distribution, allowing for the modeling and learning of DRL methods with multivariate rewards. Zhang et al. (2021) propose MD3QN, which extends distributional RL to model the joint return distribution from multiple reward sources, also aiming to learn the correlation of rewards coming from different sources.

Another area of RL which is relevant is counterfactual reasoning. Counterfactual reasoning in RL considers the outcomes of actions that were not actually taken, allowing one to ask "what if" questions about alternative decisions. By leveraging such counterfactuals, one can either augment the

data available for learning or provide more interpretable explanations of an agent's behavior. Lu et al. (2020) propose generating synthetic experience by replacing taken actions with counterfactual ones under learned dynamics, thereby improving sample efficiency. Amitai et al. (2024) instead use counterfactual trajectories to highlight how different actions would have changed the observed behavior, offering a means to interpret and communicate the agent's decision making.

# F  USED NEURAL NETWORK ARCHITECTURE

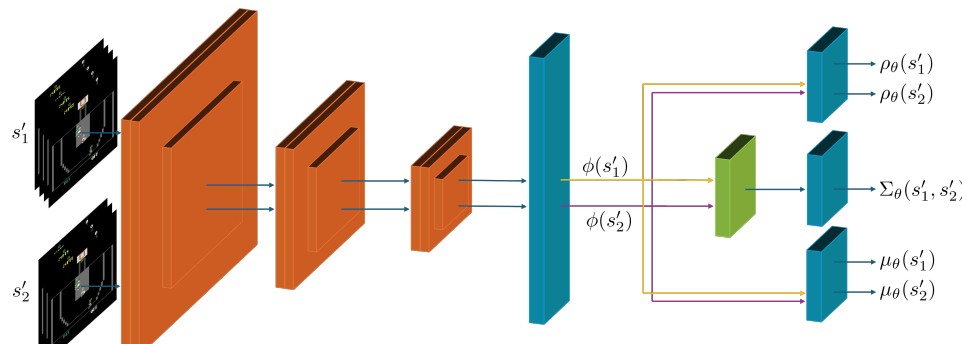

Figure 6: The architecture used in practice with Atari games from the Arcade Learning Environment. The orange blocks indicate convolutional layers. The blue blocks indicate linear layers. The first four blocks work as a feature extractor. The green block indicates an "augmentation layer". The three linear layers at the right end are, from top to bottom, the mixing, covariance, and mean heads.

We specify the architecture of the neural network, presented in Figure 6, to discuss how the $K$-GMM parameters are estimated. The general architecture follows that of DQN (Mnih et al., 2015) closely, but with a few significant differences. Firstly, the fully-connected output layer of DQN is split into three heads, estimating the mixing coefficients, the mean vectors, and the covariance matrix separately. These heads have $K$, $KN$, and $\frac{N(N+1)}{2}$ output nodes, respectively. The raw output of the mixing coefficient head is passed through the $\mathrm{softmax}$ function to produce the mixing coefficients. The output of the mean head is used directly as the estimate for the $N$-variate mean for each of the $K$ components. The output of the covariance head is used to construct a lower-triangular matrix, from which the estimate for the covariance matrix is constructed through the Cholesky composition. To ensure that the resulting estimate of the covariance matrix is positive definite, we take the exponential of its diagonal entries, and add a small positive constant.

We remark that through this process, we aim to learn a full covariance matrix with all of its off-diagonal elements, as opposed to adopting the usual assumption of diagonal covariance matrices, which would be no different than learning separate marginal distributions for each action, as in conventional DRL. It is only through the learning of these off-diagonal elements that we can prospect the interrelations and dependencies of the marginal distributions.

Furthermore, we add an *augmentation layer* before the covariance head, which takes two inputs $\mathbf{u}$, $\mathbf{v}$ and returns the vector $\begin{bmatrix}\mathbf{u} & \mathbf{v} & \mathbf{u}-\mathbf{v} & \mathbf{u}\odot\mathbf{v}\end{bmatrix}^T$, a form familiar from LSTM literature (Chen et al., 2017), which then gets input into the covariance head. In a sense, all the layers before the three heads and the augmentation layer work as a feature extractor, extracting features $\phi(s)$ from input state $s$.

The mixing coefficients for both $\eta_\theta(s; a_1, a_2)$ and $\eta_\omega^*(s_1', s_2')$ are estimated in the same manner: In the case of $\eta_\theta(s; a_1, a_2)$, a forward pass of $s$ through the network yields $\rho_\theta(s)$. In the case of $\eta_\omega^*(s_1', s_2')$, one forward pass each for $s_1'$ and $s_2'$, yield the mixing coefficients for the two univariate marginal distributions. The means follow a similar approach, where a forward pass of $s$ yields the $\mu_{\theta,k}(s)$, from which the relevant $\mu_{\theta,k}(s; a_1, a_2)$ are obtained by marginalization. Similarly, one forward pass each of $s_1'$ and $s_2'$ yield $\mu_{\omega,k}(s_1')$ and $\mu_{\omega,k}(s_2')$, from which $\mu_{\omega,k}(s_1', a_1^*)$ and $\mu_{\omega,k}(s_2', a_2^*)$ are obtained by marginalization of the optimizing action's dimension.

The estimation of the covariances follows a different pattern. $\Sigma_\theta(s)$ is obtained by a forward pass of $s$ to extract the features $\phi(s)$. Then two copies of the same $\phi(s)$ are put through the augmentation layer, resulting in the input to the covariance head being $[\phi(s) \quad \phi(s) \quad 0 \quad \phi(s) \odot \phi(s)]^T$. This input, passed through the covariance head, yields the estimate $\Sigma_\theta(s)$.

In the case of $\Sigma_\omega(s_1', s_2')$, however, the features $\phi(s_1')$ and $\phi(s_2')$ are combined in the augmentation layer to produce $[\phi(s_1') \quad \phi(s_2') \quad \phi(s_1') - \phi(s_2') \quad \phi(s_1') \odot \phi(s_2')]^T$, which yields the estimate $\Sigma_\omega(s_1', s_2')$ after passing through the covariance head.

## G  Additional Experimental Details

In the choice of loss function, left unspecified in Section 3, the Kullbeck-Leibler divergence, the Wasserstein-2 distance (Gibbs & Su, 2002), the Cramér distance (Bellemare et al., 2017b) or the Jensen-Tsallis distance (Choi et al., 2019) are all tractable candidates. All of these statistical distances between, or upper bounds thereof, are simple to obtain computationally when their arguments are two GMMs (Hershey & Olsen, 2007; Delon & Desolneux, 2020; Zhang, 2023). In this work, we choose to present results obtained with the Cramér distance, as guided by Bellemare et al. (2017b); Zhang (2023). The Cramér distance does not have a closed-form expression in the case of multivariate GMMs, so we resort to using a slicing approach guided by the Cramér-Wold theorem (An et al., 2023) as in Kolouri et al. (2020; 2018). Because the training method outlined in Section 3 involves using multiple sample transitions starting from $s$ under the same policy, it incurs some bias due to the correlation of the samples. To overcome this, we use a decaying hyperparameter $q$ which dictates that more transitions of form $\tau^2$ are used towards the beginning of training, gradually decreasing down to predominantly using transitions of form $\tau$ towards the end. This also aligns with the common MDP philosophy of explore-then-commit (Lattimore & Szepesvári, 2020), as the additional actions taken further help with exploration. For all ALE experiments, we use 3-GMMs.

## H  Broader Impact

The goal of this work to broaden the understanding of DRL to capture joint distributions of multiple actions per states. We argue for the validity of this underexplored approach, making appeal to the possible dependencies of the returns of actions at a given state, arising from dependencies in the rewards or the transition dynamics of the system. We believe there is a great deal to be explored in this area, as existing DRL algorithms have all implicitly adopted the assumption that these returns are independently distributed, or that it is of no use or interest to an agent to capture such dependencies. Although we present a concrete algorithmic method to model joint distributions of returns using GMMs, we think of these as marginal to the theoretical insights explored in the work. We believe that the methods presented in this work will serve as prototypes for further exploration in modeling joint distributions of returns in DRL, in the development of methods that have better performance, are safer, more interpretable, and better-informed.

## I  Limitations

The standard RL workflow, evidently, does not involve intentionally revisiting past states. Therefore, existing RL libraries are not suited (and furthermore, not optimized) for gathering experience replays as detailed in this work, and require heavy modification before these methods become applicable. The authors resorted to an unsophisticated and unoptimized implementation of the experience replay gathering process, which, at a state $s$, simply plays possible actions $a_1, \ldots, a_n$ one by one, observing rewards $r_1, \ldots, r_n$ and visiting next states $s_1', \ldots, s_n'$, restoring the state of the environment and the random number generator back to their previous values after each visit. We suggest, however, that in theory, it is possible to parallelize this experience gathering process, simultaneously playing the $n$ actions and observing their consequences, resulting in a great decrease in the wall-clock running time of the method.

An additional limitation of the algorithm is the number of additional hyperparameters it introduces to the training process. As stated in Appendix G, because using transition samples of form $\tau^2$ has an equivalent effect to using pairs of heavily-correlated transition samples of form $\tau$, one must

introduce the additional hyperparameter $q$ which dictates how often multivariate marginals are used in training as opposed to univariate marginals.

Furthermore, in scenarios where access to a simulator is not achievable, we posit that there may still be workarounds. For instance, Lu et al. (2020) propose the use of causal models to estimate counterfactual next state and reward outcomes for counterfactual actions. We believe such works can be a viable alternative in scenarios where we are not able to sample counterfactual outcomes through an oracle or a simulator.

More specifically, building on Example 2, we may posit that the environment's stochasticity at time $t$ stems from a shared, unobserved noise vector $U_t$. The joint outcomes would then be a function of the current state and this noise: $= f(s_t, U_t) = (S'_{t+1,a_1}, ..., S'_{t+1,a_N}, R_{t+1,a_1}, ..., R_{t+1,a_N})$. This $f$ is the true joint structural causal model (SCM). A single observed transition $(s_t, a_i, r_i, s'_i)$ is thus one marginal realization from this joint function. Then, instead of rewinding a simulation, we could collect only standard experience tuples $\tau = (s, a, r, s')$ and learn a generative model that captures $f$ given only these marginal samples. In doing so, we would learn both a generative model $G(s_t, u_t)$ (approximating $f$) and an inference network (or encoder) $E(s_t, a, r, s')$ that seeks to infer the underlying noise $\hat{u}_t$ that must have occurred to produce the observed transition. To generate a full joint transition $\tau^2$ from state $s_t$, we would (1) sample a real transition $(s_t, a, r, s')$ from the replay buffer, (2) infer the latent noise $\hat{u}_t = E(s_t, a, r, s')$, (3) generate the full set of counterfactual outcomes using this same noise vector: $(\hat{S}'_{a_1}, ..., \hat{R}_{a_N}) = G(s_t, \hat{u}_t)$. This generated $\tau^N$ tuple, or its subsamples, can then be used by our algorithm.

### I.1 EFFECT ON INFERENCE SPEED

The reduction in inference speed is negligible in practice. The main change to the model is to the size of the output layer, which now has $K + KN + \frac{N(N+1)}{2}$ nodes. The first $K$ nodes are the mixture weights for the $K$ components, the next $KN$ are the means for the $K$ components, and the remaining $\frac{N(N+1)}{2}$ constitute the entries of an $N \times N$ lower triangular matrix $L$ (due to our homoscedasticity assumption, where we assume all Gaussian components of the mixture have the same covariance matrix). We then compute the covariance matrix by the matrix product $LL^T$. There is some overhead for constructing the matrix, performing the operations on the diagonal to make sure that $LL^T$ will be positive semi-definite, and computing the matrix product $LL^T$, but none of these operations are prohibitively costly.

## J LICENSES FOR ASSETS USED

For the practical implementation of the methods described in this work, we credit the Autonomous Learning Library (Nota, 2020), whose base code repository we made extensive use of.

For the CartPole environment, we credit the OpenAI Gym library (Brockman et al., 2016). For the ALE environments, we further credit the Arcade Learning Environment (Bellemare et al., 2012). All game visuals are © Atari.

