# OpenReview forum: "Beyond Marginals: Capturing Dependent Returns through Joint Moments in Distributional Reinforcement Learning"
_ICLR.cc/2026/Conference — Submitted to ICLR 2026_

### Official Review · Reviewer_7hCi · 2025-10-27

**Soundness:** 3
**Presentation:** 3
**Contribution:** 2
**Rating:** 4
**Confidence:** 5

**Summary:**

This paper proposes modeling the joint return distribution in distributional reinforcement learning. The authors formalize this problem by constructing a joint MDP and deriving the corresponding joint Bellman equations and policy evaluation schemes. Building on this theoretical framework, they introduce an algorithm that represents the joint return distribution using a deep Gaussian Mixture Model (GMM). The proposed approach is then evaluated by numerical experiments.

**Strengths:**

* The paper is overall well-written and easy to follow.
* The theoretical derivation is clear and sound.
* The idea of considering joint return distribution is intuitive.

**Weaknesses:**

* Most importantly, the method relies on access to a $\tau^2$ trajectory, an assumption that is incompatible with the standard RL framework. Although the author argues such a trajectory can be obtained by assuming an omniscient simulator (like digital twins), this is not the case for nearly all known practical applications. This core assumption severely limits the practical scope of the paper.
* The experiments are all done in environments where $N$ is small. The authors should discuss whether the proposed approach can scale up as $|\mathcal{S}|$ and $|\mathcal{A}|$ become large.
* Section 4.2 appears to contain an error. The authors define a safe control objective $\max_\pi \pi^\top\mu-\lambda\pi^\top\Sigma\pi$ and call it a quadratic program. Since $\mu$ and $\Sigma$ depend on $\pi$, this is not a QP. This formulation needs to be corrected or clarified.

**Questions:**

* The paper appears to primarily leverage the first and second-order moments of the joint return distribution for the downstream analysis. Could the authors clarify the necessity of modeling the entire distribution? What specific advantages does this full-distributional approach offer over a simpler method that might directly model only the first two moments?
* The framework's benefits for the interpretability of MDPs are clear. However, could the authors elaborate on its utility for decision-making? Specifically, does modeling the joint return distribution enable policy improvements or optimization strategies that are not achievable with existing methods (e.g., standard distributional RL or moment-based approaches)?

---

> ### Author Response · Authors · 2025-11-22
> **Response to Reviewer 7hCi [1/2]**
>
> ## Part (1/2)
>
> ### Weaknesses
>
> **"Most importantly, the method relies on access to a trajectory, an assumption that is incompatible with the standard RL framework. Although the author argues such a trajectory can be obtained by assuming an omniscient simulator (like digital twins), this is not the case for nearly all known practical applications. This core assumption severely limits the practical scope of the paper."**
>
> We kindly refer the reviewer to item 1 in the general response.
>
> **"The experiments are all done in environments where $N$ is small. The authors should discuss whether the proposed approach can scale up as $|\mathcal{S}|$ and $|\mathcal{A}|$ become large."**
>
> **In DRL literature, the commonly used benchmarks are ALE environments, in which every environment has at most $|\mathcal{A}|=18$ actions. (See, for instance, [1, 2, 3, 4], etc.)** As discussed in item 2 of the general response, the number of parameters of the mixture of Gaussians scales polynomially in the number of actions (quadratically in the second moment components and linearly in the first moment.) This is far better than exponential growth, which would follow for instance from a naive extension of categorical distribution to the multivariate setting. Furthermore, a natural and intuitive extension of the proposed method to continuous action spaces also exists in the form of mixtures of Gaussian processes. The only tangible difference of this setting to the one already proposed would be that $\mu$ and $\Sigma$ would now be functions of $(s, a) \in \mathcal{S}\times\mathcal{A}$, where $\mathcal{A}$ is a continuous domain, rather than a finite set. Concretely, we define a stochastic process, $Z_s$, indexed by the action $a \in \mathcal{A}$. The mean function of this process is the standard Q-function, $\mu_s(a) = \mathbb{E}[Z(s,a)]$, which existing methods seek to approximate. The covariance function (or kernel) of this process would be $k_s(a_i, a_j) = \text{Cov}(Z(s, a_i), Z(s, a_j))$. Our approach would learn this kernel $k_s$. Instead of learning $N^2$ discrete covariance terms, the algorithm would learn the hyperparameters of the kernel function. A benefit of this formulation is that it avoids $\mathcal{O}(N^2)$ scaling: the complexity is merely determined by the parameterization of the kernel, not the (infinite) number of actions.
>
> **"Section 4.2 appears to contain an error. The authors define a safe control objective $\max_\pi \pi^T\mu - \lambda \pi^T\Sigma\pi$ and call it a quadratic program. Since $\mu$ and $\Sigma$ depend on $\pi$, this is not a QP. This formulation needs to be corrected or clarified."**
>
> We kindly refer the reviewer to item 3 of the general response.
>
> ### Questions
>
> **"The paper appears to primarily leverage the first and second-order moments of the joint return distribution for the downstream analysis. Could the authors clarify the necessity of modeling the entire distribution? What specific advantages does this full-distributional approach offer over a simpler method that might directly model only the first two moments?"**
>
> We do model the entire distribution, but this model is already parameterized by the statistics we estimate, which are the first two moments of each mixture component (plus, a set of mixture weights). **Therefore, the entire distribution already follows with practically no extra cost from the moments of the components. The choice of using a mixture is for increase in the expressivity of the model.** We refer the reviewer to item 2 in the general response for a deeper discussion adjacent to this.

---

> ### Author Response · Authors · 2025-11-22
> **Response to Reviewer 7hCi [2/2]**
>
> ## Part (2/2)
>
> **"The framework's benefits for the interpretability of MDPs are clear. However, could the authors elaborate on its utility for decision-making? Specifically, does modeling the joint return distribution enable policy improvements or optimization strategies that are not achievable with existing methods (e.g., standard distributional RL or moment-based approaches)?"**
>
> **As we showcase in the paper, modeling the joint return distribution helps not only with interpretability but also in safety and cost-efficiency.** We have provided explicit formulations where the use of the use of, for instance, the second-order statistics leads to improvement in secondary considerations, in Sections 4.2, 4.3, and 4.4, respectively. **In addition, joint modeling can also influence the training dynamics in a beneficial way. A well-known advantage of distributional RL over conventional value-based RL (e.g., DQN) is that the learning signal is enriched: instead of regressing to a single scalar target, the agent learns from a full distributional loss, which typically provides a denser and more informative gradient signal.
> Analogously, in our setting, moving from univariate to multivariate return distributions similarly enriches the training objective. By comparing full multivariate distributions rather than just marginal (univariate) ones, the loss now reflects discrepancies in joint structure, such as correlations and higher-order dependencies, that were previously ignored. We expect this multivariate distributional loss to serve as a more informative and potentially more effective training signal.**
>
> ### References
>
> [1] Bellemare, Marc G., Will Dabney, and Rémi Munos. "A distributional perspective on reinforcement learning." International conference on machine learning. PMLR, 2017.
>
> [2] Dabney, Will, et al. "Distributional reinforcement learning with quantile regression." Proceedings of the AAAI conference on artificial intelligence. Vol. 32. No. 1. 2018.
>
> [3] Dabney, Will, et al. "Implicit quantile networks for distributional reinforcement learning." International conference on machine learning. PMLR, 2018.
>
> [4] Nguyen-Tang, Thanh, Sunil Gupta, and Svetha Venkatesh. "Distributional reinforcement learning via moment matching." Proceedings of the AAAI conference on artificial intelligence. Vol. 35. No. 10. 2021.

---

### Official Review · Reviewer_XMpF · 2025-10-27

**Soundness:** 3
**Presentation:** 1
**Contribution:** 2
**Rating:** 2
**Confidence:** 3

**Summary:**

1. The motivation is clear. Modeling the joint distribution among actions is natural and technically sound. It would be well expected that the additional correlation information among actions would be beneficial for policy optimization or the general decision-making.

2. Experiments are comprehensive, which involves multiple domains. I found it interesting to establish the connection between the correlation information among actions with the interpretability of sequential decision-making.

**Strengths:**

1.The motivation is clear. Modeling the joint distribution among actions is natural and technically sound. It would be well expected that the additional correlation information among actions would be benefits for policy optimization or the general decision making.

2.Experiments are comprehensive, which involves multiple domains. I found it is interesting to establish the connection between the correlation information among actions with the interpretability of sequential decision making.

**Weaknesses:**

1. **Paper organization and writing need to be largely improved**. (1) The empirical results are provided very early, which is too far away from the experimental sections, undermining the readability. (2) I think the motivation of a joint modeling of return distribution among actions is easy to understand. Therefore, there is no need to list two naïve examples in the introduction part. It would be more suggested to directly focus on the general setting with a clear illustration in the introduction. (3) It is not clear whether it is necessary to propose the POMDP framework, which seems less connected with the practical algorithmic design and following theoretical analysis. (4) Jointly learning the correlations of actions is not a novel idea, which is natural in the policy gradient literature. However, this idea is not widely adopted, as additionally learning the correlation, such as the covariance matrix, often increases the computational instability. Therefore, this paper should focus on discussion of this practical factors. What is the price we need to pay for practical algorithms?

2. **Theoretical results are less convincing**. (1) Some formulations and explanations are on binary action settings, e,.g., in Line 209, and it is less clear how to extend them to the general action case. (2) This paper implicitly applies the Gaussian assumption or is restricted to the first and second moments of learning in the joint distribution learning. This limitation should be elaborated. What is the consequence? (3) In Section 2.2, it is less clear how the joint Bellman operator works on both the first and second moments of return distribution. (4) If we only consider the first two moments, the results can also be extended to Wasserstein distance in Theorem 1. In Theorems 2 and 3, can we derive the contraction property of the joint Bellman operator?

3. **Limited novelty of algorithm**. (1) Using the mixture Gaussian modeling is limited in novelty. (2) In line 346, does the summation increase exponentially in terms of the state space?

3. **Explanations of experimental results are less satisfactory**.  In my opinion, the experiments should focus on large-scaled setting with more Atari games and Mujoco environments. The policy evaluation setting and Cliff walking are weak. This is because readers will easily expect the benefits of using joint distribution modeling in practice. They expect some off-the-shelf methods to further improve the existing (distributional) RL algorithms. I think the explanations in the section 4.3 look interesting, and relevant explanations on more experiments are expected. This would also contribute to interpretable RL.

**Questions:**

Overall, I think this research direction is potential, but the current version of the paper fails to sufficiently present the research value with the critical weaknesses limited above. Here are my other questions of this paper.

1.Figure 1 is introduced in the first section, but there lacks sufficient explanation, e.g., about the benefits of a joint learning. It is more suggested to use a more general and clear illustration in the intro with more detailed explanations.

2.It is hard to follow the motivation of POMDP. More explanations are helpful.

3.In line 177, we estimate the true parameters where $\hat{\mu}_1$ and so on are the estimates instead of the true parameters.

4.How do we do the inference when we have the covariance matrix among actions? The drawback is that it reduces the inference speed in practice. This limitation also needs to be emphasized.

---

> ### Author Response · Authors · 2025-11-21
> **Response to Reviewer XMpF [1/5]**
>
> ## Part(1/5)
>
> ### Weaknesses
>
> **"The empirical results are provided very early, which is too far away from the experimental sections, undermining the readability."**
>
> The abundance of experiments in the paper have led to the necessity for including many tables and figures. Because of this, some numerical results had to be placed in earlier pages, so that no single page would be consist only of tables and figures. We have tried to improve the placement in the revision, by moving all tables and figures forward.
>
> **"I think the motivation of a joint modeling of return distribution among actions is easy to understand. Therefore, there is no need to list two naïve examples in the introduction part. It would be more suggested to directly focus on the general setting with a clear illustration in the introduction."**
>
> We appreciate the feedback. **We thought it necessary to display the two possible sources of dependencies in MDPs, namely, those arising from the reward function and transition dynamics.** In our view, it is important to clearly illustrate these with intuitive examples, because it is natural that the first reaction to the claim that MDPs may possess such dependence structures is denial along with claiming that these would violate the definition of an MDP.
>
> **"It is not clear whether it is necessary to propose the POMDP framework, which seems less connected with the practical algorithmic design and following theoretical analysis."**
>
> In a similar vein to the previous response, we believe that the POMDP view is important to show that one can indeed formally and rigorously instantiate MDPs with a joint reward / transition structure. **This formalism lets us view any standard MDP as allowing us a view into this joint POMDP in the background. Although, as the reviewer correctly points out, this framework is less connected to the following theoretical analysis, the entire premise of the analysis in fact depends on the (joint) oracle access as defined in lines 207-211 of the original manuscript.**
>
> **"Jointly learning the correlations of actions is not a novel idea, which is natural in the policy gradient literature. However, this idea is not widely adopted, as additionally learning the correlation, such as the covariance matrix, often increases the computational instability. Therefore, this paper should focus on discussion of this practical factors. What is the price we need to pay for practical algorithms?"**
>
> Firstly, we would like to kindly remind that we are in the value-based setting rather than the policy-based setting. **To our knowledge, learning joint return distributions in the value-based setting has never been explored and is entirely novel.** We would appreciate it greatly if the reviewer could cite some relevant papers that learn these joint distributions / dependence structures in the policy-based setting so that we can give them appropriate reference in our work. From our inquiry, the works we have found are akin to [1], where the authors propose modeling the stochastic policy as a mixture of Gaussian distributions. This is entirely different to our approach, where we model the distributions of returns as mixtures of Gaussian distributions. **In particular, these works operate in continuous action spaces and assume the policy is a distribution, such as a multivariate Gaussian $\pi(a|s) = \mathcal{N}(\mu_s, \Sigma_s)$, where $\Sigma_s$ is a parameter of the actor that describes the agent's own stochasticity. Their goal is still to improve the optimization of the standard $\max_\pi \mathbb{E}[Z^\pi]$ objective. But, ours is a property of the environment's underlying stochastic transition and reward functions, describing the statistical dependency of counterfactual outcomes.**
>
> We do not believe that the computational instability factor applies to our setting, as evidenced by the deep learning results shown in Section 4.4, where the training process convincingly finds near-optimal policies with relatively low SEM values. As for the most significant practical factor, we consider it to be the reliance on counterfactual outcomes, a deeper discussion of which we have provided in the first item of the general response.

---

> ### Author Response · Authors · 2025-11-21
> **Response to Reviewer XMpF [2/5]**
>
> ## Part (2/5)
>
> **"Theoretical results are less convincing. (1) Some formulations and explanations are on binary action settings, e,.g., in Line 209, and it is less clear how to extend them to the general action case."**
>
> We would like to make sure to defuse any potential misunderstanding about the setting. **We do not make the assumption that the action space is binary.** Because we choose to model our return distributions as mixtures of Gaussian distributions, it suffices for us to only consider first and second moments, as these statistics are sufficient to parameterize such distributions. In the reviewer's example, line 209 is formalizing access to joint transitions $\tau^2$ which give us access to the next states and rewards attained from two different actions. **In our case, this is enough since second moments relate information about *pairs* of actions, no matter what the cardinality of the action space is.** If one wants to model using distributions that require higher-order moments, it is not hard to generalize the results of the paper to this setting.  For instance, for third moments, we would indeed require samples that give us information about *triples* of actions. We would then have to assume oracle access from the POMDP that gives us this kind of information, i.e., we would need $\Omega := \mathcal{S}^3 \times \mathbb{R}^3$ and $O(o \mid x, a_1):= \delta_{(s_{a_1}, s_{a_2}, s_{a_3}),(r_{a_1}, r_{a_2}, r_{a_3})}(o)$. It is straightforward from this example to see how this would generalize to yet higher-order moments. We have also added this discussion to the manuscript as a remark.
>
> **"(2) This paper implicitly applies the Gaussian assumption or is restricted to the first and second moments of learning in the joint distribution learning. This limitation should be elaborated. What is the consequence?"**
>
> We kindly refer the reviewer to item 2 in the general response.
>
> **"(3) In Section 2.2, it is less clear how the joint Bellman operator works on both the first and second moments of return distribution."**
>
> In Section 2.2, we represent the joint Bellman equations of Proposition 1 in the form of an operator, very similar to how it is done in the usual policy evaluation scheme in conventional RL with the Bellman equations and the Bellman (consistency) operator. Starting from an arbitrary initialization $M_0$, which is a state- and action-index collection of $(N+1)$ scalars, the first scalar representing an estimate for the expected return of $(s, a)$, the second representing an estimate for the variance of the return of $(s, a)$, and the remaining $(N-1)$ representing estimates for the cross-covariances of the return of action $a$ at state $s$ with the $(N-1)$ other actions that can be taken at $s$, all under the evaluated policy.
>
> The proposed scheme updates its estimates $M_t$ by repeatedly applying the joint Bellman operator, i.e., setting $M_{t+1} \leftarrow \mathcal{T}_{2, N}^\pi M_t$, where the operator updates the estimate $M_t$ as the right-hand sides of the joint Bellman equations of Proposition 1.
>
> Theorem 1 then shows that this iterative evaluation scheme converges to the mean and second moments of the returns for each $(s,a)$ with a linear rate. We would be happy to provide further clarification if needed.
>
> **"(4) If we only consider the first two moments, the results can also be extended to Wasserstein distance in Theorem 1."**
>
> If the reviewer means that one can extend the convergence results to the distributional convergence of a sequence of Gaussian distributions parameterized by the JIPE-estimated iterates $\mu^k$, $\Sigma^k$ to the Gaussian distribution parameterized by the true mean and covariance $\mu$, $\Sigma$ of a policy $\pi$ in Wasserstein distance, this is precisely true and this is the result we present in Theorem 6 of our Appendix B.
>
> **"In Theorems 2 and 3, can we derive the contraction property of the joint Bellman operator?"**
>
> We do not define the joint Bellman operator as an operator on distributions but on the moments of the distributions, therefore it would not be applicable to ask whether the contraction property can be proved in Theorems 2 and 3, which deal with distributions. We prove the distributional convergence of GMMs with an alternative argument compared to a fixed-point iteration. We refer the reviewer to Appendix C for the proofs of these results in the Cramér distance, and to Appendix B for the analogous results in the Wasserstein-2 distance.

---

> ### Author Response · Authors · 2025-11-21
> **Response to Reviewer XMpF [3/5]**
>
> ## Part (3/5)
>
>
> **"Limited novelty of algorithm. (1) Using the mixture Gaussian modeling is limited in novelty."**
>
> We do not claim novelty of using Gaussian mixtures for modeling. **The GMM is only used as a tool for demonstrating *the actual novelty of the proposed method, which is the modeling of cross-action return dependencies.*** Furthermore, we do acknowledge in our work that there exists previous literature that models the distribution of returns with a mixture of Gaussians, but as we also point out, these works only deal with marginal, one-dimensional distributions of returns, whereas we propose using multivariate Gaussian components for the first time.
>
> **"(2) In line 346, does the summation increase exponentially in terms of the state space?"**
>
> It does not, as the summation is over mixture components and is not related to the size of the state space. The summation in line 346 gives us the TD-target distribution, which is a Gaussian mixture with $K^2$ components. We then look at the distributional distance (in practice, the Cramér distance) of our current estimate of the return distribution, which is a Gaussian mixture with $K$ components. This distance is then used and backpropagated, after which SGD is used to update the parameters of the neural network which estimates the return distributions.
>
> **"Explanations of experimental results are less satisfactory. In my opinion, the experiments should focus on large-scaled setting with more Atari games and Mujoco environments. The policy evaluation setting and Cliff walking are weak."**
>
> As also mentioned in item 1 of the general response, **we see the contribution of the work as more on the theory side, as foundational in highlighting this previously unexplored corner of joint modeling of returns in RL. We would like to point out that we have included ALE environments in our work to show that this approach also remains viable in richer and more complex settings.** Because experiments on these more complex environments over millions of training frames take considerable time, we will add results on several more ALE environments in a later revision, but we also believe this should not shift the focus from the main contribution of the paper. Similarly, the policy evaluation environments and Cliff Walking were deliberately designed to be simple, intended for justifying and providing intuition about the theory in the work. Here, we would also like to restate all of the experiments present in the paper:
>
> 1. The Windy Gridworld and Shared-Randomness Bandit examples that are intended to coroborate the JIPE algorithm,
>
> 2. The Cliff Walking experiments to showcase the use of our methods in the context of safety.
>
> However, we also have a set of application-oriented experiments using very standard benchmark RL environments such as Atari games. Namely, we have experiments on
>
> 1. The CartPole and Pong environments, in the context of interpretability
>
> 2. The ALE environments Atlantis, Boxing, and Pong in the context of performance and cost-efficiency.

---

> ### Author Response · Authors · 2025-11-22
> **Response to Reviewer XMpF [4/5]**
>
> ## Part (4/5)
>
> **"This is because readers will easily expect the benefits of using joint distribution modeling in practice. They expect some off-the-shelf methods to further improve the existing (distributional) RL algorithms."**
>
> We respectfully disagree with the core premise that the benefits of our proposed joint-distributional modeling would be "easily expected" by readers. On the contrary, in optimization and reinforcement learning, demonstrating that modeling a distribution provides tangible benefits for an expected-value objective is often a non-obvious and significant research contribution in itself. We would like to draw an analogy to stochastic optimization. When minimizing an expected loss $\min \mathbb{E}[L(w)]$, the stochastic gradient descent (SGD) algorithm is already asymptotically optimal for this objective. It is not obvious that a benefit exists from modeling the full distribution of the SGD iterates. The objective is to optimize the expectation, and the algorithm for doing so does not require this distributional knowledge. An even more direct analogy exists within our own field. Before the advent of DRL, traditional RL methods (like Q-learning) were already designed to be optimal for the standard $\max_\pi \mathbb{E}[Z^\pi]$ objective. It was a foundational and non-obvious discovery by [2] that modeling the full return distribution (e.g., C51) could practically stabilize learning and lead to superior convergence for that same expected-return objective. This benefit was not easily expected; it had to be rigorously demonstrated and has since launched an entire sub-field. Our paper follows this exact scientific pattern. The reviewer states that readers "expect... methods to further improve existing (distributional) RL algorithms," but demonstrating this non-obvious improvement is precisely our contribution. **It is not obvious that modeling the joint-action return covariance $\Sigma(s)$, a property no existing DRL algorithm captures nor proposed to study, should provide any benefit. Our paper is the first to pose this novel question, propose a formal framework and tractable algorithm (JIPE) to model this new information, and demonstrate that this new information (the covariance) does, in fact, unlock entirely new, practical capabilities in safety, interpretability, and cost-efficiency. These benefits are non-trivial and, like the original DRL papers, required the extensive theoretical justification and empirical demonstration that our paper provides.**
>
> **"I think the explanations in the section 4.3 look interesting, and relevant explanations on more experiments are expected. This would also contribute to interpretable RL."**
>
> We appreciate the positive feedback. We will look into extending the section on interpretability with more examples.
>
> ### Questions
>
> **"Figure 1 is introduced in the first section, but there lacks sufficient explanation, e.g., about the benefits of a joint learning. It is more suggested to use a more general and clear illustration in the intro with more detailed explanations."**
>
> Given the space constraint, we were forced to keep the caption of Figure 1 more concise but please note that the interpretation is provided in the beginning of Section 4.3.
>
> **"It is hard to follow the motivation of POMDP. More explanations are helpful."**
>
> We have included an explanation that clarifies the POMDP formalism with the assumption of an oracle access to joint transitions, as detailed in our previous answers. To reiterate, the POMDP formalism lets us view any standard MDP as allowing us a view into this unobserved joint POMDP.
>
>
> **"In line 177, we estimate the true parameters where $\hat{\mu}_1$ and so on are the estimates instead of the true parameters."**
>
> We see what the reviewer means and agree that the mentioned sentence is not clear. We intended that sentence to mean: "We use the sample mean and variance (to make the estimations) $\hat{\mu_1}, \hat{\mu_2}, \hat{\sigma}^2_1, \hat{\sigma}^2_2$ (to the true mean and variances)." We have clarified this sentence in the revision.

---

> ### Author Response · Authors · 2025-11-22
> **Response to Reviewer XMpF [5/5]**
>
> ## Part (5/5)
>
> **"How do we do the inference when we have the covariance matrix among actions? The drawback is that it reduces the inference speed in practice. This limitation also needs to be emphasized."**
>
> The reduction in inference speed is negligible in practice. The main change to the model is to the size of the output layer, which now has $K + KN+ \frac{N(N+1)}{2}$ nodes. The first $K$ nodes are the mixture weights for the $K$ components, the next $KN$ are the means for the $K$ components, and the remaining $\frac{N(N+1)}{2}$ constitute the entries of an $N\times N$ lower triangular matrix $L$ (due to our homoscedasticity assumption, where we assume all Gaussian components of the mixture have the same covariance matrix). We then compute the covariance matrix by $LL^T$. There is some overhead for constructing the matrix, performing some operations on the diagonal to make sure that $LL^T$ will be PSD, and computing the matrix product $LL^T$, but none of these operations are prohibitively costly. We have added a discussion of this to Appendix I.
>
> ### References
>
> [1] Ciosek, Kamil, and Shimon Whiteson. "Expected policy gradients for reinforcement learning." Journal of Machine Learning Research 21.52 (2020): 1-51.
>
> [2] Bellemare, Marc G., Will Dabney, and Rémi Munos. "A distributional perspective on reinforcement learning." International conference on machine learning. PMLR, 2017.

---

### Official Review · Reviewer_v334 · 2025-10-29

**Soundness:** 2
**Presentation:** 2
**Contribution:** 2
**Rating:** 2
**Confidence:** 4

**Summary:**

This paper considers the problem of outcome (reward and next state) dependence among different actions in reinforcement learning (RL). To estimate the joint distribution of returns of all actions at a state, the paper models a POMDP, named the joint MDP, and designs the joint iterative policy evaluation (JIPE) algorithm.
The authors derive the joint distributional Bellman equations, analyze their contractive properties, and thereby establish the convergence of the JIPE algorithm. For practical implementation, the paper proposes using a Gaussian mixture model to model the joint distribution, and employs neural networks to estimate the mean function and covariance function.
Finally, the authors conduct extensive experiments to demonstrate the effectiveness of their proposed method.

**Strengths:**

1. The authors clearly illustrate the problem they address. For instance, Figure 1 provides an intuitive visualization of the outcome dependence phenomenon, which effectively helps readers understand the problem.

2. The paper presents error analysis. Additionally, the authors conduct a detailed analysis of their experimental results.

**Weaknesses:**

1. The paper’s motivation lacks clarity: While I acknowledge that the phenomenon of outcome dependence is natural in RL, the authors fail to clearly elaborate on why this problem is critical.
From an algorithmic perspective, learning an optimal policy does not necessarily require accounting for such outcome dependence. Although the authors attempt to demonstrate the superiority of their proposed JIPE algorithm across various problems in the experimental section, these demonstrations are not sufficiently convincing, as detailed below:​

In Section 4.2, the authors consider the cliff walking problem and attempt to show that accounting for outcome dependence can yield a risk-averse policy. However, they do not explain how to obtain the optimal solution to the proposed Markowitz problem—this problem is, in fact, highly similar to the mean-variance optimization problem discussed in Chapter 7 of Bellemare et al. (2023) and is notoriously difficult to solve. Furthermore, the experimental results only briefly report mean, standard deviation, maximum, and minimum, without including common risk measures (e.g., Value-at-Risk, Conditional Value-at-Risk). Additionally, risk-sensitive RL algorithms are absent from the baselines in this set of experiments.

In Sections 4.3 and 4.4, the authors use examples such as Cartpole and Pong to argue that the effective rank of the covariance matrix can distinguish whether the agent is in a "critical state". They claim that it can help reduce operational costs when action execution incurs costs. While this analysis seems intuitively appealing, the argument is incomplete and unconvincing. For instance, the authors do not rule out the possibility that the value functions (i.e. expected returns) of different actions are close in "post-critical states"; in such cases, using only first-order information (value function) to decide whether to act would be simpler and more justifiable. I recommend that the authors refine this part and provide more rigorous reasoning to support their claims.

2. The proposed algorithm relies on a simulator capable of counterfactual reasoning, which imposes significant limitations on its practical applicability.

**Questions:**

see weakness

---

> ### Author Response · Authors · 2025-11-21
> **Response to Reviewer v334 [1/2]**
>
> ## Part (1/2)
>
> ### Weaknesses
>
> **"The paper’s motivation lacks clarity: While I acknowledge that the phenomenon of outcome dependence is natural in RL, the authors fail to clearly elaborate on why this problem is critical. From an algorithmic perspective, learning an optimal policy does not necessarily require accounting for such outcome dependence."**
>
> A designation of which problem is "critical" is highly subjective, and so is the designation of which policy is "optimal". **We outline several settings where the learning of these joint distributions lead to observable improvement in multiple or secondary considerations. Namely, these are interpretability, safety and cost-efficiency.** They can very well be considered critical if a decision-maker values these considerations in addition to optimizing the expected returns, and the importance of such secondary considerations is established in many application areas of RL, one example being autonomous driving. We do not "require" to learn these statistics if our only consideration for optimality is maximizing the expected return, but *such a view rules out the entirety of distributional reinforcement learning*, because one does not need to learn the whole distribution of returns in order to find policies whose returns are optimal in expectation, as learning the expected value of returns is sufficient in that case.
>
> **"Although the authors attempt to demonstrate the superiority of their proposed JIPE algorithm across various problems in the experimental section, these demonstrations are not sufficiently convincing, as detailed below:
> In Section 4.2, the authors consider the cliff walking problem and attempt to show that accounting for outcome dependence can yield a risk-averse policy. However, they do not explain how to obtain the optimal solution to the proposed Markowitz problem—this problem is, in fact, highly similar to the mean-variance optimization problem discussed in Chapter 7 of Bellemare et al. (2023) and is notoriously difficult to solve. "**
>
> We kindly refer the reviewer to item 3 in the general response.
>
> **"Furthermore, the experimental results only briefly report mean, standard deviation, maximum, and minimum, without including common risk measures (e.g., Value-at-Risk, Conditional Value-at-Risk)."**
>
> We appreciate the reviewer's suggestion. We have included the CVaR values at thresholds $0.1$ and $0.05$ in the updated manuscript.
>
> **"Additionally, risk-sensitive RL algorithms are absent from the baselines in this set of experiments."**
>
> We would like to once again state, as we have in the general response, that **we view the main contribution of the paper to be the highlighting of the previously unexplored joint return distribution perspective in DRL, and the theoretical contributions that follow from this perspective, such as the theoretically-backed joint Bellman operator and the JIPE scheme.** The purpose of Sections 4.2 through 4.4 are merely to showcase possible use cases where the learned joint statistics might be leveraged and the new structure we are revealing could potentially be exploited. We do not claim to have revolutionized risk-averse RL and these results are not meant to be taken as the main highlight of the work, beating all previous benchmarks. For this reason, we do not see comparison against algorithms who were specifically designed from the get-go to be risk-averse as necessary or fair, and we believe that such results might even mislead readers.

---

> ### Author Response · Authors · 2025-11-21
> **Response to Reviewer v334 [2/2]**
>
> ## Part (2/2)
>
> **"In Sections 4.3 and 4.4, the authors use examples such as Cartpole and Pong to argue that the effective rank of the covariance matrix can distinguish whether the agent is in a "critical state". They claim that it can help reduce operational costs when action execution incurs costs. While this analysis seems intuitively appealing, the argument is incomplete and unconvincing. For instance, the authors do not rule out the possibility that the value functions (i.e. expected returns) of different actions are close in "post-critical states"; in such cases, using only first-order information (value function) to decide whether to act would be simpler and more justifiable. I recommend that the authors refine this part and provide more rigorous reasoning to support their claims."**
>
>
> We sincerely appreciate the feedback. This is indeed an important point raised by the reviewer. **We would like to highlight that value-function indifference is not the same as outcome irrelevance.** The value function (mean) can dangerously obscure the underlying risk structure as demonstrated by the following example: Consider a state $s$ where $\mu(s, a_1) = \mu(s, a_2) = 0.5$. An agent solely considering the first moment sees indifference and might simply play NOOP to save on cost. Let us consider two cases: **(i)** $Z(s, a_1) = Z(s, a_2) = 0.5$ with probability $1$. Here, $\sigma^2_1 = \sigma_2^2 = 0$,  $\rho_{12} = +1.0$ and $\text{erank}(\Sigma) = 1$. The actions are truly irrelevant and playing NOOP is a valid choice. **(ii)** $Z(s, a_1)$ is $0$ or $1$ with equal probability. $Z(s, a_2) =  Z(s, a_1)$ with probability $\epsilon$ and $Z(s, a_2) = 1- Z(s, a_1)$ with $1-\epsilon$. Here, $\sigma^2_1 = \sigma^2_2=1/4$, $\rho_{12} = \frac{1}{4}(1-2\epsilon)$. $\text{erank}(\Sigma)=\exp(-(1-\epsilon)\log(1-\epsilon)-\epsilon\log\epsilon$ is a continuous function of $\epsilon$, taking every value in $[0, 1]$ as $\epsilon$ sweeps from $0$ to $1/2$. Then, for any choice of $\delta \in (0, 1)$, an $\epsilon$ may be chosen such that $\delta < \text{erank}(\Sigma) <1-\delta$, deeming the state critical. **A first-order agent cannot discriminate between these two cases, but our method, by using $\text{erank}(\Sigma)$, can.** We have added this example to the manuscript.
>
> **"The proposed algorithm relies on a simulator capable of counterfactual reasoning, which imposes significant limitations on its practical applicability."**
>
> We kindly refer the reviewer to item 1 in the general response.

---

### Official Review · Reviewer_693N · 2025-10-31

**Soundness:** 3
**Presentation:** 3
**Contribution:** 3
**Rating:** 8
**Confidence:** 3

**Summary:**

* The main contribution of the paper is to highlight the importance of joint return modeling in Distributional Reinforcement Learning (DRL). It proposes to explicitly model the coupling between the returns of different actions at a given state using K-component Gaussian Mixture Models (GMMs). By introducing a joint MDP formulated as a POMDP with hidden coupled potential outcomes, the paper derives joint distributional Bellman relations and, in particular, introduces a Joint Iterative Policy Evaluation (JIPE) method that comes with convergence guarantees for joint means and second moments.

* The experimental section focuses on synthetic MDPs and small-scale control tasks to validate the theoretical results and to qualitatively demonstrate some benefits of the proposed approach, namely interpretability (through alignment with state criticality), safety (via improved control of the mean/variance trade-off), and cost-effectiveness.

**Strengths:**

* The paper presents a principled and mathematically rigorous formulation, supported by solid proofs including convergence guarantees. The problem is clearly specified, particularly through the joint MDP/POMDP formulation with joint transitions. Provided that a simulator or digital-twin environment is available—capable of counterfactual replay of potential outcomes (state transitions and rewards)—the estimation of cross-moments is feasible.

* Although the experiments are limited to synthetic MDPs and relatively simple “toy” problems, they convincingly validate the theoretical framework and effectively illustrate the potential advantages of the proposed method.

**Weaknesses:**

* The assumption that one can counterfactually replay next-state/reward outcomes for other actions at the same state may be infeasible in many real-world scenarios. It would be valuable to provide practical insights or strategies for mitigating this limitation.

* The modeling of the joint return distribution is restricted to second-order statistics (means, variances, and covariances). The authors could strengthen their discussion by justifying why this second-order approximation is often sufficient, or by outlining possible extensions to higher-order moments and non-Gaussian distributions.

* Similarly, it would be helpful to discuss potential extensions of the proposed approach to continuous action spaces.

**Questions:**

(No question in particular. See previous section for suggestions of improvements.

---

> ### Author Response · Authors · 2025-11-21
> **Response to Reviewer 693N**
>
> ### Strengths
>
> **"Although the experiments are limited to synthetic MDPs and relatively simple “toy” problems, they convincingly validate the theoretical framework and effectively illustrate the potential advantages of the proposed method."**
>
> We would like to point out that we do have synthetic MDPs deliberately designed to be simple "toy" problems, intended for justifying and providing intuition about the theory in the work. Namely, these are:
>
> 1. The Windy Gridworld and Shared-Randomness Bandit examples that are intended to coroborate the JIPE algorithm,
>
> 2. The Cliff Walking experiments to showcase the use of our methods in the context of safety.
>
> **However, we also have a set of application-oriented experiments using very standard benchmark RL environments such as Atari games.** Namely, we have experiments on
>
> 1. The CartPole and Pong environments, in the context of interpretability
>
> 2. The ALE environments Atlantis, Boxing, and Pong in the context of performance and cost-efficiency,
>
> These last set of experiments support our message that learned joint statistics can very well be extended to richer and more complex settings.
>
> ### Weaknesses
>
> **"The assumption that one can counterfactually replay next-state/reward outcomes for other actions at the same state may be infeasible in many real-world scenarios. It would be valuable to provide practical insights or strategies for mitigating this limitation."**
>
> We kindly refer the reviewer to item 1 in the general response.
>
> **"The modeling of the joint return distribution is restricted to second-order statistics (means, variances, and covariances). The authors could strengthen their discussion by justifying why this second-order approximation is often sufficient, or by outlining possible extensions to higher-order moments and non-Gaussian distributions.
> "**
>
> We kindly refer the reviewer to item 2 in the general response.
>
> **"Similarly, it would be helpful to discuss potential extensions of the proposed approach to continuous action spaces."**
>
> This is a great direction for an extension, we thank the reviewer for the suggestion. **The most natural, intuitive extension of the proposed approach to continuous action spaces would be to use a mixture of Gaussian processes** instead of mixture of Gaussian distributions. The only tangible difference of this setting to the one already proposed would be that $\mu$ and $\Sigma$ would now be functions of $(s, a) \in \mathcal{S}\times\mathcal{A}$, where $\mathcal{A}$ is a continuous domain, rather than a finite set. Concretely, we define a stochastic process, $Z_s$, indexed by the action $a \in \mathcal{A}$. The mean function of this process is the standard Q-function, $\mu_s(a) = \mathbb{E}[Z(s,a)]$, which existing methods seek to approximate. The covariance function (or kernel) of this process would be $k_s(a_i, a_j) = \text{Cov}(Z(s, a_i), Z(s, a_j))$. Our approach would learn this kernel $k_s$. Instead of learning $N^2$ discrete covariance terms, the algorithm would learn the hyperparameters of the kernel function. A benefit of this formulation is that it avoids $\mathcal{O}(N^2)$ scaling: the complexity is merely determined by the parameterization of the kernel, not the (infinite) number of actions.
> **We do note, however, that in the DRL literature, it is commonplace to assume a discrete action space, as can be observed for instance in [1, 2, 3, 4], and the like.**
>
> ### References
>
> [1] Bellemare, Marc G., Will Dabney, and Rémi Munos. "A distributional perspective on reinforcement learning." International conference on machine learning. PMLR, 2017.
>
> [2] Dabney, Will, et al. "Distributional reinforcement learning with quantile regression." Proceedings of the AAAI conference on artificial intelligence. Vol. 32. No. 1. 2018.
>
> [3] Dabney, Will, et al. "Implicit quantile networks for distributional reinforcement learning." International conference on machine learning. PMLR, 2018.
>
> [4] Nguyen-Tang, Thanh, Sunil Gupta, and Svetha Venkatesh. "Distributional reinforcement learning via moment matching." Proceedings of the AAAI conference on artificial intelligence. Vol. 35. No. 10. 2021.

---

### Author Response · Authors · 2025-11-21
**General Response [1/3]**

## Part (1/3)

We thank the reviewers for their kind praise of the work and their constructive criticism. We are honored to see that the reviewers remarked that the work is well-written and easy to follow **(Reviewer 7hCi)**, principled and mathematically rigorous, supported by solid, clear and sound theory **(Reviewers 693N, 7hCi)**, that the potential advantages of the proposed method are effectively illustrated **(Reviewer 693N)**, that the experimental results are comprehensive **(Reviewer XMpF)** and analyzed in detail **(Reviewer v334)**, that the motivation is clear and the main interest of the paper is natural, intuitive and technically sound **(Reviewers 7hCi, XMpF)**. We believe that a work that simultaneously possesses all of these qualities is worthy of publication in ICLR.

In the following, we would like to address some of the common points raised collectively by the reviewers.

1. **The method is based on counterfactual next-state/reward outcomes for multiple actions at the same state (what is stylized as $\tau^2$ trajectories in the manuscript). This may be unattainable in real-world scenarios and some practical applications. (Reviewers 693N, v334, 7hCi)**

We would like to remind the reviewers that **the main contribution of this paper is meant to be theoretical**. Our main goal is to draw attention to this unexplored perspective on reinforcement learning and primarily attempt to lay its theoretical foundations. We envision that follow-up works can potentially leverage the ideas presented here in a more application-oriented, practically-polished manner. With this being said, it is not out of the ordinary to see, in theoretical analyses pertaining to reinforcement learning, to presume access to a generative model, which one can query at will like an oracle to obtain samples of the reward or the next state of a state-action pair [1, 2, 3]. We believe it is not far-fetched to make a similar assumption on the existence of a generative model which would instead give us joint trajectories $\tau^2$. In fact, **this is essentially what is being assumed in lines 207-211 of the original manuscript.** There already exist published works that assume such access in explainable RL literature, cf. [4] for one example.

Still, although one can ideally assume any premise to draw theoretical conclusions, we are aware that such conclusions are of little worth if the assumptions are almost sure not to hold in practice. For this reason, we also wanted to provide **practical scenarios where it is straightforward to implement such an oracle access. As can be read in lines 212-230 of the original manuscript, in common benchmark simulation environments that virtually all RL works use, such as the ALE Atari environments or Gymnasium control environments, it is remarkably simple to achieve this with only a handful lines of code changed.** It is as easy as saving a temporary backup of the state of the random number generator(s) and the environment before taking an action, recording the next state and reward that follow, restoring the state of the random number generator(s) and the environment to the backed-up states, taking a different action and recording its next state, and rewards, and so on, for the required number of joint actions. Because all such rewards/next states are obtained under the same, common source of stochasticity, we then view these as joint samples from the joint POMDP. We invite the reviewers to view our implementation within the supplementary files we have shared along with our original submission.

**We do acknowledge that there will naturally be scenarios and applications where we will not be able to achieve this, or have a simulator at all. Even in these scenarios, we posit that there are possible workarounds.** For instance, we refer the reviewers to [5], which proposes the use of causal models to estimate counterfactual next state/reward outcomes for actions alternative to the one taken. We believe such works can be a viable alternative in scenarios where we are not able to sample counterfactual outcomes through an oracle or a simulator.

---

### Author Response · Authors · 2025-11-21
**General Response [2/3]**

## Part (2/3)

More specifically, building on Example 2 from our submission, we posit that the environment's stochasticity at time $t$ stems from a shared, unobserved noise vector $U_t$. The joint outcomes (next states and rewards for all actions) are a function of the current state and this noise: $(S'\_{t+1, a\_1}, ..., S'\_{t+1, a\_N}, R\_{t+1, a\_1}, ..., R\_{t+1, a\_N}) = f(s\_t, U\_t)$. This $f$ is the true joint structural causal model (SCM). A single observed transition $(s\_t, a\_i, r\_i, s'\_{i})$ is thus one marginal realization from this joint function. Then, instead of rewinding, we would collect only standard experience tuples $\tau = (s, a, r, s')$ and learn a generative model that captures $f$ given only these marginal samples. In doing so, we would learn both a generative model $G(s\_t, u\_t)$ (approximating $f$) and an inference network (or encoder) $E(s\_t, a, r, s')$ that seeks to infer the underlying noise $\hat{u}\_t$ that must have occurred to produce the observed transition. To generate a full joint transition $\tau^2$ from state $s\_t$, we would (1) sample a real transition $(s\_t, a, r, s')$ from the replay buffer, (2) infer the latent noise $\hat{u}\_t = E(s\_t, a, r, s')$, (3) generate the full set of counterfactual outcomes using this same noise vector: $(\hat{S}'\_{a\_1}, ..., \hat{R}\_{a\_N}) = G(s\_t, \hat{u}\_t)$. This generated $\tau^N$ tuple can then be used by our algorithm.

We have added remarks on this topic to Appendix I of the updated manuscript to address any concerns related to the matter.

2. **Why are we restricted to the first and second moments / Why do we assume "that the distribution is Gaussian?" (Reviewers 693N, XMpF, 7hCi)**

We would like to first emphasize that our modeling of the return distributions in Section 3 is not by Gaussian distributions, but by **mixtures of Gaussian distributions**. This is a critical distinction. Gaussian distributions are limited in expressivity and come with plenty of implicit restrictions about the shape of the distribution: That it is unimodal, symmetric, etc., which customarily do not hold for return distributions in many RL scenarios of interest (see Section 5.4 of [2] for a discussion). Gaussian mixture models, on the other hand, are much more expressive and lift many such restrictions. Indeed, it is well known that **finite Gaussian mixtures are dense in the space of continuous probability distributions under standard metrics, i.e., a Gaussian mixture model with enough (but finitely many) components can approximate any continuous distribution arbitrarily well.** Our Theorems 5 and 7, in the appendix, also highlight this fact in the context of RL, under the Wasserstein and Cramér distances.

The sufficiency of the second-order approximation follows as a design choice. Any DRL algorithm has to make a choice on how they want to model the return distributions and this consequentially dictates what they need to estimate to parameterize their model. **In our case, we decide to model the distributions as GMMs, and second-order statistics are sufficient to do this.** However, ideally, a smart design choice would involve choosing a model which can then be exploited favorably for the considerations that the designer wants to perform well in. **In our manuscript, we have purposefully highlighted numerous concrete applications for which first-order statistics (the means) are not sufficient, but the second-order statistics are. Namely, these applications are safety, interpretability, and cost-efficiency.** We have provided explicit formulations where the use of these second-order statistics leads to improvement in secondary considerations, in Sections 4.2, 4.3, and 4.4, respectively. It is reasonably straightforward to extend the proposed framework to even higher-order moments, such as skewness or kurtosis, with minimal changes. Notably, these higher order statistics would require joint samples involving more counterfactual actions, i.e., trajectories of form $\tau^3 = (s, a\_1, a\_2, a\_3, r\_1, r\_2, r\_3, s'\_1, s'\_2, s'\_3, a\_1', a\_2', a\_3')$, and so on.

Finally, an additional factor in favor of using Gaussian mixtures to model joint distributions is the fact that **many of the more granular distributions that are used in the marginal, univariate setting (such as the categorical distribution of C51) scale exponentially in the number of parameters with the dimensionality of the distribution, whereas the number of parameters of Gaussian distributions scale only polynomially.** Additionally, quantile (QR-DQN-like) methods are not easily extendable to our joint setting, as a multivariate quantile function is not uniquely, tractably defined for optimization, and does not offer a simple regression target.

---

### Author Response · Authors · 2025-11-21
**General Response [3/3]**

## Part (3/3)

3. **The Markowitz problem of Section 4.2 is erroneous / not a quadratic program / solving it is near infeasible in practice. (Reviewers v334, 7hCi)**

We have realized that at least two reviewers have some questions about the Markowitz problem of Section 4.2, and both of these ultimately tie back to the same source of confusion. Firstly, this is a valid quadratic program, and the quantities $\mu(s)$ and $\Sigma(s)$ at each state $s$ are independent of the optimization variable $\pi$ and are functions of the unsafe policy $\pi_u$. As we state in the manuscript, $\mu$ and $\Sigma$ follow from the evaluation (by JIPE) of the unsafe policy $\pi_u$. They thus constitute the state-indexed collection of mean vectors and covariance matrices of returns belonging to policy $\pi_u$. We subsequently solve the quadratic program $\max_{\pi\in\Delta_N} \pi^T\mu(s) -\lambda\pi^T\Sigma(s)\pi$ at each state $s$, and we use a solution $\pi_M$ of this problem as our deployment policy. **This process can alternatively be viewed as a post-hoc one-step policy improvement step, which, given relevant (first- and second-order) statistics of a policy $\pi_u$, returns a policy $\pi_M$, improved in the sense of safety.** We have taken the general confusion as feedback that this part should be clarified and better worded, and made the necessary updates.

### References

[1] Agarwal, Alekh, et al. "Reinforcement learning: Theory and algorithms." CS Dept., UW Seattle, Seattle, WA, USA, Tech. Rep 32 (2019): 96.

[2] Bellemare, Marc G., Will Dabney, and Mark Rowland. Distributional reinforcement learning. MIT Press, 2023.

[3] Kearns, Michael, and Satinder Singh. "Finite-sample convergence rates for Q-learning and indirect algorithms." Advances in neural information processing systems 11 (1998).

[4] Amitai, Yotam, Yael Septon, and Ofra Amir. "Explaining reinforcement learning agents through counterfactual action outcomes." Proceedings of the AAAI Conference on Artificial Intelligence. Vol. 38. No. 9. 2024.

[5] Lu, Chaochao, et al. "Sample-efficient reinforcement learning via counterfactual-based data augmentation." arXiv preprint arXiv:2012.09092 (2020).

---

### Author Response · Authors · 2025-12-02
**Message to AC**

Dear Area Chair,

Thank you for handling our submission *“Beyond Marginals: Capturing Dependent Returns through Joint Moments in Distributional Reinforcement Learning.”* Reviewer 693N recommends acceptance, calling the work principled, mathematically rigorous, and supported by convincing experiments. The other reviewers also find the theory sound and interesting. Their reservations focus on perceived practicality and on questions about how to interpret and deploy our framework, not on flaws in the core results.

We summarize the main points raised and how we addressed them, with pointers to our detailed rebuttal, which we hope you will consult for technical details and precise references.

1.	**Access to joint counterfactual trajectories/simulators.**
Reviewers question the realism of assuming joint counterfactual outcomes. In our rebuttal to Reviewers XMpF and v334, we clarify that this is a theoretical paper and that oracle/generative-model assumptions are standard in RL theory. In common simulators (Atari, Gym control) joint trajectories are straightforward to obtain via RNG control and checkpointing. We also point to causal model based methods as a route beyond simulators **(item 1 of our General Response Parts 1 and 2/3).**

2.	**Gaussianity and sufficiency of second-order statistics.**
Some reviewers interpret our approach as assuming Gaussian returns. In our rebuttal to Reviewers 693N and XMpF, we emphasize that we use Gaussian mixtures, which are universal approximators. In our parameterization, learning first and second moments per component is enough to reconstruct the full joint distribution. The restriction to joint moments is a deliberate, pragmatic choice that yields concrete benefits (safety, interpretability, cost-efficiency), and we explain how the framework extends to higher-order moments when more joint samples are available **(item 2 of our General Response Part 2/3, and the response to Reviewer 693N on the extension to mixtures of Gaussian processes).**

3.	**Safety / Markowitz formulation and risk-sensitive baselines.**
Two reviewers question whether our Markowitz-style problem is a valid QP and whether we compare fairly to risk-sensitive methods. We clarify that the mean vector and covariance matrix are computed under a fixed “unsafe” policy. The optimization variable is only the deployment policy, making the per-state problem a standard quadratic program and interpretable as a covariance-aware policy improvement step. We also added CVaR metrics and additional discussion to show that joint moments can systematically produce safer policies, rather than aiming to beat every specialized risk-averse baseline **(item 3 of our General Response Part 3/3 and the response to Reviewer v334 on the absence of risk-sensitive RL algorithms from the baselines).**

4.	**Motivation, experimental scope, interpretability, safety and cost-efficiency.**
Reviewer v334 questions whether outcome dependence is “critical,” v334 and XMpF ask for broader empirical validation. We provide concrete examples where identical marginal value functions but different covariance structure lead to very different risk profiles, and we show how joint modeling enables: **(i)** separating value-indifference from outcome irrelevance, **(ii)** safer control, and **(iii)** interpretability. We stress that, for a theory-focused paper, our experimental suite, consisting of controlled finite MDPs, CartPole, and five Atari games, is fully in line with the empirical scope of ICLR theory papers and is not a weakness of the work **(the response to Reviewer 693N and the response to Reviewer v334 Part 1/2 for the experimental design and scope).**

5.	**Scalability, continuous actions, and technical clarity.**
Concerns about scaling in $|A|, |S|$ and continuous actions are addressed in our rebuttal to XMpF and v334. We show that GMM parameter counts grow only polynomially and outline an extension to mixtures of Gaussian processes for continuous actions, whose complexity depends on the GP parameterization, not action space cardinality. We also clarified the role of the POMDP formulation, the joint Bellman operator on moments, and the convergence of joint GMMs **(the response to Reviewer 7hCi Part 1/2 for scalability and complexity and the response to Reviewer XMpF Parts 1 and 2/5 for the POMDP formulation and convergence).**

**This paper makes a non-incremental move from marginal DRL to a joint DRL perspective, introducing new theoretical machinery** (joint MDP/POMDP, joint Bellman relations, JIPE, convergence of joint GMMs). **The reviews largely agree on the novelty and soundness of this theory. We respectfully submit that, in line with the ICLR guidelines on high-risk, less-explored directions, the work should be evaluated primarily on these conceptual and theoretical contributions, with our empirical evaluation, which we believe is sufficiently polished and extensive for a theory paper, supporting, rather than limiting, its impact.**

---

### Meta-Review · Area_Chair_nSRG · 2026-01-12

**Summary:**

This paper proposes an ambitious and well-motivated extension of distributional reinforcement learning by arguing that modeling return distributions independently across actions neglects important dependency structure induced by shared dynamics and rewards. The authors introduce a joint MDP/POMDP formulation, derive joint distributional Bellman equations, propose the Joint Iterative Policy Evaluation (JIPE) scheme with convergence guarantees, and demonstrate how joint moments can be leveraged for interpretability, safety, and cost-aware policy deployment. The work is mathematically rigorous and explores a novel conceptual direction that is underdeveloped in the current DRL literature.

Despite these strengths, I do not believe the paper is yet ready for acceptance at ICLR. Multiple reviewers raised concerns about the realism and scope of the assumptions—most notably access to joint counterfactual outcomes—as well as about the practical impact of modeling joint second-order statistics alone. While the rebuttal clarifies the authors’ intent and defends the theoretical framing, some concerns remain only partially resolved, particularly regarding applicability beyond simulator-based settings and the extent to which the empirical results convincingly demonstrate advantages over marginal DRL approaches. I view this work as promising and potentially impactful, but it would benefit from further consolidation, either through stronger empirical evidence or a tighter reframing as a foundational theory paper with explicitly bounded scope. I encourage the authors to pursue a revised submission.

**Reviewer Concerns:**

Several reviewer concerns were addressed by the rebuttal. Reviewer 693N's questions regarding the restriction to second-order statistics and the expressiveness of the proposed modeling approach were addressed through clarification of the use of Gaussian mixture models and discussion of extensions to higher-order moments. Concerns raised by 7hCi  about the Markowitz-style optimization and its interpretation were addressed by clarifying that the optimization is performed as a post-hoc deployment step based on fixed policy statistics, making it a well-defined quadratic program. Across reviewers, the authors also clarified the theoretical motivation, convergence guarantees, and the intended role of the experiments.

Some concerns remain outstanding. Reviewer v334 expressed skepticism about whether outcome dependence is sufficiently critical in practice to justify the added complexity, and whether the empirical results clearly demonstrate benefits over marginal DRL baselines; these concerns were only partially alleviated. Reviewer XMpF raised concerns about the practicality of assuming access to joint counterfactual trajectories outside simulators, and while the rebuttal discusses oracle and causal-model-based workarounds, this assumption remains a significant limitation. These unresolved issues around realism and empirical decisiveness ultimately motivate my recommendation.

**Reviewer Scores:**

Reviewer 693N.
This reviewer was strongly positive and emphasized the soundness and conceptual novelty of the work. I expect 693N would have maintained their score after discussion.

Reviewer 7hCi.
This reviewer was generally positive but sought clarification on optimization details and interpretation. Given the rebuttal, I expect 7hCi would have maintained their score or increased it slightly.

Reviewer v334.
This reviewer raised substantive concerns about practical relevance and empirical justification. While some technical points were clarified, I expect v334 would have maintained a low score or increased it only marginally.

Reviewer XMpF.
This reviewer remained skeptical about the realism of the assumptions and deployment beyond simulators. The rebuttal clarifies intent but does not fully resolve these concerns. I expect XMpF would have maintained their score after discussion.

Overall, discussion clarified the scope and intent of the contribution but is unlikely to have produced consensus, with remaining disagreement reflecting differing expectations about applicability and maturity rather than correctness.

---

### Decision · Program_Chairs · 2026-01-26

Reject